# Rapid-charging aluminium-sulfur batteries operated at 85 °C with a quaternary molten salt electrolyte

Jiashen Meng[1,2], Xufeng Hong[1], Zhitong Xiao[1], Linhan Xu[1], Lujun Zhu[1], Yongfeng Jia[1], Fang Liu[2], Liqiang Mai ®[2] ✉ & Quanquan Pang ®[1] ✉

Molten salt aluminum-sulfur batteries are based exclusively on resourcefully sustainable materials, and are promising for large-scale energy storage owed to their high-rate capability and moderate energy density; but the operating temperature is still high, prohibiting their applications. Here we report a rapid-charging aluminium-sulfur battery operated at a sub-water-boiling temperature of 85 °C with a tamed quaternary molten salt electrolyte. The quaternary alkali chloroaluminate melt – possessing abundant electrochemically active high-order Al-Cl clusters and yet exhibiting a low melting point – facilitates fast $Al^{3+}$ desolation. A nitrogen-functionalized porous carbon further mediates the sulfur reaction, enabling the battery with rapid-charging capability and excellent cycling stability with 85.4% capacity retention over 1400 cycles at a charging rate of 1 C. Importantly, we demonstrate that the asymmetric sulfur reaction mechanism that involves formation of polysulfide intermediates, as revealed by *operando* X-ray absorption spectroscopy, accounts for the high reaction kinetics at such temperature wherein the thermal management can be greatly simplified by using water as the heating media.

Large-scale electrochemical energy storage technologies are gaining increasing global attention as we continue to promote the penetration of sustainable energy[1–3]. Currently, lithium-ion batteries (LIBs) that use flammable organic electrolytes are dominant largely owed to the reduced manufacturing cost as the production scales[4,5]. However, the high materials cost, non-sustainable lithium resource and safety concerns with LIBs are predicted to hinder their wide penetration. Rechargeable aluminum batteries (RABs) are attractive as the alternative owed to the high abundance, low cost, and high capacity of aluminum metal (2.98 Ah g$^{-1}$ and 8.05 Ah cm$^{-3}$)[6–8]. Sulfur has found its use as the cathode for RABs owed to its high capacity and low cost[9,10].

However, the Al−S batteries have suffered poor rate capability and cycling stability[11]. Efforts have been devoted to promoting the rate capability and stability of Al−S battery[12–14]. As with other metal−sulfur batteries, one approach was to construct sophisticated carbon−sulfur composite cathodes to enhance the sulfur conversion kinetics[15–17]. Another was to regulate the electrolyte to lower the $Al^{3+}$ desovlation barrier and thus to improve the overall reaction kinetics of an Al−S battery[18,19]. Salient efforts on modifying the ionic liquid electrolytes (1-ethyl-3-methylimidazolium chloride with AlCl₃, EMIC-AlCl₃) by doping with Br$^-$ or Li$^+$ have proved effective to promote $Al^{3+}$ desolvation. Inorganic molten salt electrolytes have led to dramatic improvement in the reaction kinetics of RABs, in addition to their advantages of low cost and inflammability[20]. We recently reported on fast-charging aluminum−chalcogen batteries by using alkali chloroaluminate melt electrolytes[21]. The Al−Se (and Al−S) batteries showed excellent fast-charging capability at an operation temperature of 180 °C (and 110 °C). Nevertheless, maintaining such temperature may require sophisticated sealing and thermal management systems, and

[1]Beijing Key Laboratory of Theory and Technology for Advanced Batteries Materials, School of Materials Science and Engineering, Peking University, 100871 Beijing, China. [2]State Key Laboratory of Advanced Technology for Materials Synthesis and Processing, School of Materials Science and Engineering, Wuhan University of Technology, 430070 Wuhan, China. ✉e-mail: mlq518@whut.edu.cn; qqpang@pku.edu.cn

restrict its mobile applications. However, if we can reduce the operation temperature to below 100 °C, water can be used as a low-cost heating media, and the battery start-up heating will be less energy-intensive for frequent on-and-off encountered in mobile operation. The thermal management system can be greatly simplified, as well as the sealing and thermal-insulating materials, in a way that the proton exchange membrane fuel cells are used.

Molten salts have been granted with a sign of overly high temperature 200–600 °C and being impractical for batteries. In this work, we break the convention and present a resourcefully sustainable molten salt Al−S battery operated at as low as 85 °C, which simultaneously realizes fast reaction kinetics and high cycling stability. The quaternary alkali chloroaluminate melt electrolyte shows a lower melting point of -80 °C than the conventional binary and ternary systems, while maintaining fast diffusion kinetics and rapid Al deposition/stripping. Our experimental analyses and theoretical calculations confirm the presence of abundant high-order $Al_nCl_{3n+1}^-$ clusters in the quaternary alkali chloroaluminate melt even at a low temperature of 85 °C, which results in a low $Al^{3+}$-desolvation barrier, laying the ground for fast cell charging. We further show a high-surface-area carbon framework with abundant nitrogen sites is chemically compatible with the melt electrolyte and further accelerates the sulfur conversion reaction. The demonstrated Al−S battery presents a high capacity of 931 mAh g$^{-1}$ with a small voltage hysteresis (0.19 V) at a charging rate of $C$/5, and shows excellent high-rate cycling stability over 1400 cycles at 1 C. This work opens up possibilities for practical applications of sustainable Al−S batteries in both static and mobile energy storage with intrinsic safety and cost-effectiveness.

## Results

### Quaternary alkali chloroaluminate melt electrolyte

Inorganic molten salts are known as low-cost and high-activity electrolytes and have been widely used in metal electrodeposition, high-temperature batteries, silicon refining and electro-splitting of $CO_2$[22–24].

Typical molten salts exhibit high melting point of up to 600 °C and strong corrosive feature, and therefore these electrochemical devices require a high operation temperature and sophisticated anti-corrosion sealing, leading to high energy consumption and high manufacturing cost. It is desired to develop low-melting-point molten salts that not only enables a low operation temperature but also guarantees reversible operation of the electrochemical devices. In our work, we propose a quaternary alkali chloroaluminate melt based on an AlCl$_3$−NaCl−LiCl−KCl mixture, which has a lower melting point of -80 °C than the binary and ternary molten salt systems (Fig. 1a)[25,26]. The alkali chloroaluminate melts consist AlCl$_3$ as Lewis acids and alkali metal chlorides as Lewis bases. When changing these ratios away from the low eutectic point, their melting points will increase. Therefore, the salt ratio of the melt electrolytes as an important parameter has great influence on their physical properties (e.g., melting point, viscosity and ionic conductivity), and directly determines the battery electrochemical performance. For molten salts, the latent heat of fusion dictates the energy required to overcome the electrostatic attraction forces between cations and anions; a weaker electrostatic attraction force leads to a lower fusion temperature (Fig. 1b)[27]. From the perspective of thermodynamics, in our quaternary melt, the presence of multiple components greatly increases the fusion entropy, leading to decreased fusion temperature. Furthermore, the higher degree of cation disordering and increased complexity of $Al_nCl_{3n+1}^-$ clustering anions (as discussed below) further reduce the degree of configuration symmetry among the ions, lowering the electrostatic attraction forces and thus the melting point[28,29]. The low melting point of -80 °C for the quaternary alkali melt is foundational for the low temperature and yet fast operation of our molten salt Al−S batteries. Besides, the coexisting alkali metal cations (e.g., Li$^+$, Na$^+$, and K$^+$) not only keep electric neutrality of molten salts, but also distribute around $Al_nCl_{3n+1}^-$ clusters to breakdown the "net-work" structure by their asymmetric interactions, which are beneficial for lowering the viscosity and increasing the ionic conductivity.

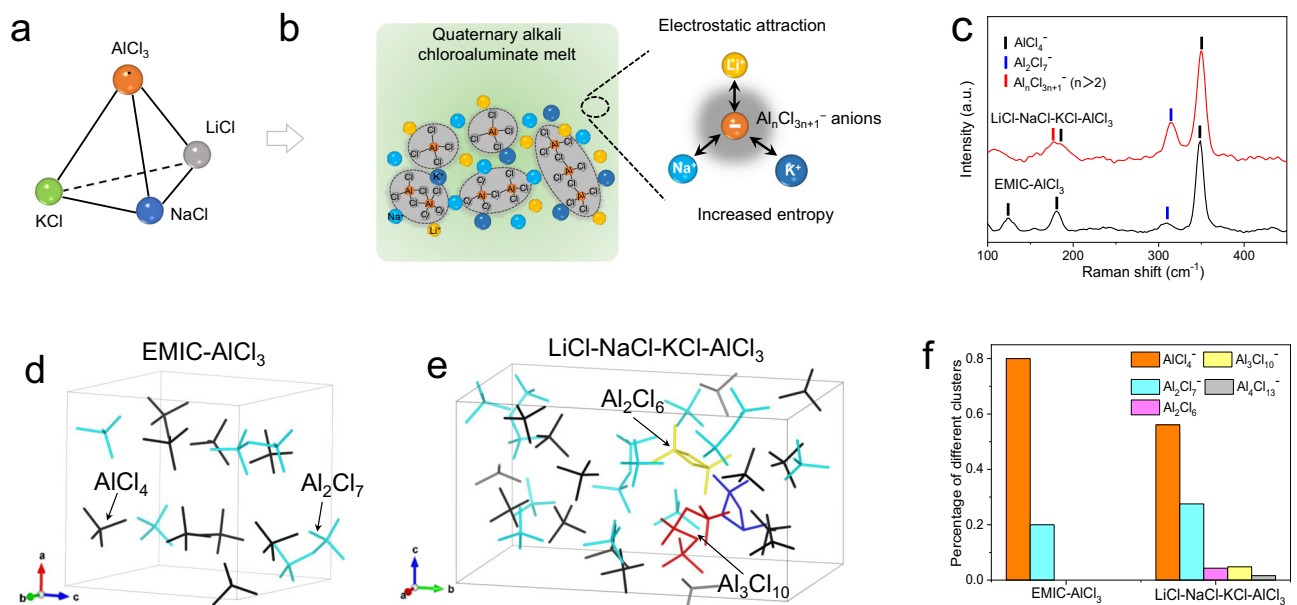

**Fig. 1 | Design principles and structural characterizations of the quaternary alkali chloroaluminate melt electrolyte. a** Tetrahedral diagram of the four components for inorganic molten salt electrolyte, showing the basis for the phase diagram for the electrolyte formulations. **b** Design principles for the quaternary alkali melt electrolyte, showing the presence of high-order $Al_nCl_{3n+1}^-$ cluster that has a more delocalized charge density. **c** Raman spectra of the quaternary alkali chloroaluminate melt (AlCl$_3$−NaCl−LiCl−KCl) and ionic liquid (EMIC-AlCl$_3$)

electrolytes. **d**, **e** Typical snapshots from the AIMD trajectories in equilibrium showing the different solvation clusters for the organic ionic liquid (**d**) and quaternary alkali melt electrolytes (**e**). The $AlCl_4^-$, $Al_2Cl_7^-$, $Al_2Cl_6$, and $Al_3Cl_{10}^-$ clusters are shown in black, cyan, yellow, and red. The cations Li$^+$, Na$^+$, and K$^+$ are omitted for clarity. **f** Percentage of different Al−Cl clusters in the quaternary alkali melt and ionic liquid electrolytes, as quantified from AIMD trajectories.

Further, the $Al^{3+}$ desolvation kinetics in chloroaluminate melt electrolytes is correlated with the form of the $Al_nCl_{3n+1}^-$ clusters, which further depends on the temperature[30]. By means of theoretical simulations, we have previously proposed that the fast Al desolvation kinetics at 180 °C is closely related to the lower Al–Cl bond dissociation barriers in higher-order $Al_nCl_{3n+1}^-$ clusters for the binary NaCl–AlCl₃ molten salt electrolyte[21]. The ionic conductivity of the quaternary melt electrolyte was investigated using electrochemical impedance spectroscopy. By analyzing the Nyquist plot, the ionic conductivity is calculated to be around 0.66 S m⁻¹, guaranteeing fast reaction kinetics (Supplementary Fig. 1). We now investigate the solvation structure and desolvation kinetics of $Al^{3+}$ at the lower operation temperature of 85 °C for the quaternary melt. Raman spectroscopy was first performed to identify the form of $Al_nCl_{3n+1}^-$ clusters in the electrolyte by comparing with the organic ionic liquid electrolyte (EMIC–AlCl₃). The two typical peaks located at 310.7 and 349.2 cm⁻¹ correspond to the $Al_2Cl_7^-$ and $AlCl_4^-$ signals, respectively (Fig. 1c)[31]. The intensity ratio of the two peaks is much higher in the alkali chloroaluminate melt than that in the organic ionic liquid, indicating a higher fraction of $Al_2Cl_7^-$ which is known as the major electrochemically active cluster for organic ionic liquids. Furthermore, a new shoulder peak at around 175.3 cm⁻¹ appears for the alkali chloroaluminate melt, which is attributed to higher-order clusters ($Al_nCl_{3n+1}^-$, $n > 2$)[32].

To further reveal the solvation structure around $Al^{3+}$ of the quaternary alkali chloroaluminate melt at 85 °C, ab initio molecular dynamics (AIMD) simulations were carried out (detailed in Supplementary Movies 1 and 2). In the EMIC–AlCl₃ ionic liquid, only $AlCl_4^-$ and $Al_2Cl_7^-$ clusters are observed (Fig. 1d), and indeed, the

Al plating/stripping in ionic liquids are based on the conversion between $AlCl_4^-$ and $Al_2Cl_7^-$ [33]. On basis of statistical analyses of the AIMD trajectory, the molar ratio of $AlCl_4^-$ and $Al_2Cl_7^-$ is close to 4 (Fig. 1f). In striking contrast, the quaternary alkali chloroaluminate melt shows the presence of high-order $Al_3Cl_{10}^-$ along with neutral $Al_2Cl_6$, which have demonstrated to be more energetically favorable for $Al^{3+}$ desolvation based on a multi-step desolvation reaction (Fig. 1e)[21]. Statistical analyses further show the presence of an even more desolvation-favorable $Al_4Cl_{13}^-$ cluster in equilibrium, and that the molar ratio of $AlCl_4^-$ and $Al_2Cl_7^-$ decreases to -2 (Fig. 1f), which is consistent with the Raman spectroscopy studies above. Further, with this evidence we also hypothesize that the formation of high-order polyatomic clusters ($Al_3Cl_{10}^-$ and $Al_4Cl_{13}^-$)—that possess more delocalized charge density and hence are less charge-polarizable—accounts for the lowered melting point of the quaternary melt; this ought to be fundamentally driven by increased configuration entropy of the alkali cations. Indeed, the calculated charge density differences of the $Al_nCl_{3n+1}^-$ ($n > 2$) clusters show that much less charge density is concentrated on the longer bridging Al–Cl bonds, which facilitates the Al–Cl bond dissociation and the release of free $Al^{3+}$ (Supplementary Fig. 2).

To investigate the reversibility of Al plating/stripping process in the quaternary melt at 85 °C, an Al|Mo cell was assembled. Aluminum undergoes a highly reversible stripping and plating with a low overpotential (Fig. 2a). The desolvation reaction kinetics were evaluated by the exchange current measurement of Al|Al symmetric cells using linear scanning voltammetry. The cell in the alkali chloroaluminate melt exhibits a higher exchange current density of 0.195 mA cm⁻² than that in the ionic liquid (0.145 mA cm⁻²), indicating faster desolvation

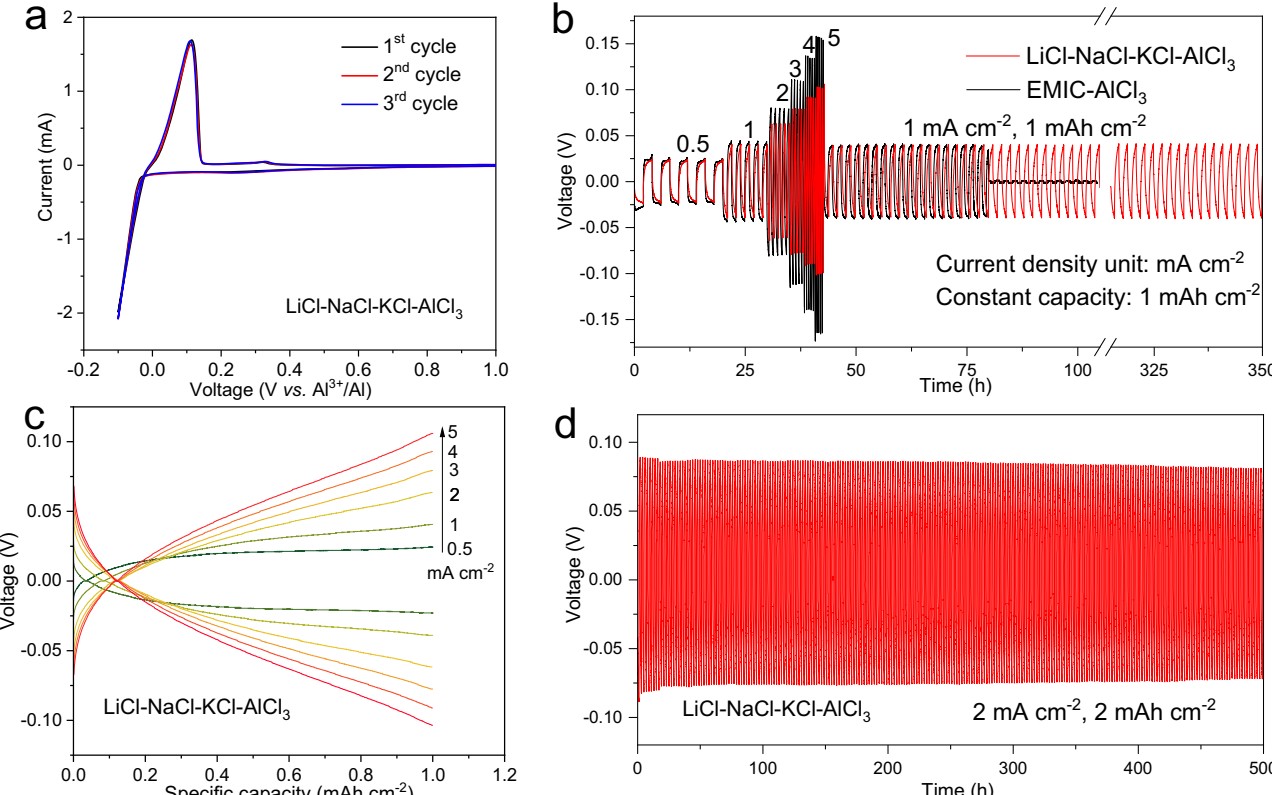

**Fig. 2 | Electrochemical evaluation of the quaternary alkali chloroaluminate melt and ionic liquid electrolytes at 85 °C. a** CV plots of the Al|Mo cell using quaternary alkali melt at a scan rate of 5 mV s⁻¹ in the home-made Swagelok cell schematically shown in the inset of (**a**). **b** Rate performance of Al|Al symmetric cells using quaternary alkali melt and ionic liquid electrolytes at various current densities from 0.5 to 5 mA cm⁻² with a constant striping/plating capacity of 1 mAh cm⁻². **c** The corresponding voltage profiles of the Al|Al symmetric cell at varied current density using the quaternary melt electrolyte. **d** Cycling performance of Al|Al symmetric cell using the quaternary melt electrolyte at a current density of 2 mA cm⁻² with a constant capacity of 2 mAh cm⁻².

kinetics (Supplementary Fig. 3a, b). The reaction kinetics were further investigated by galvanostatic cycling of the Al|Al symmetric cells at 85 °C. The rate performances were characterized by cycling the cells at various current densities with a fixed striping/plating capacity of 1.0 mAh cm⁻². The cell fitted with the quaternary melt electrolyte exhibits a voltage polarization of 47, 80, 125, 157, 184, and 210 mV at a current density of 0.5, 1.0, 2.0, 3.0, 4.0, and 5.0 mA cm⁻², respectively (Fig. 2b). These values are, respectively, lower than those of the cell using organic ionic liquid electrolyte (Fig. 2c and Supplementary Fig. 3c). The molten salt-based cell can sustain stable cycling for over 350 h at 1.0 mA cm⁻², while a sudden voltage drop occurred in the cell fitted with ionic liquid electrolyte at 78 h of operation, which is attributed to dendrite induced short-circuit. This further confirms the dendrite suppression effect in the quaternary melt electrolyte. Further, at a higher current density of 2 mA cm⁻² with a capacity of 2 mAh cm⁻², the quaternary melt-based symmetric cell exhibits good cycling stability for over 250 cycles with a low voltage polarization of approximately 160 mV (Fig. 2d and Supplementary Fig. 3d). Therefore, the quaternary melt electrolyte enables fast Al³⁺ desolation kinetics even at a low temperature of 85 °C, thus resulting in low voltage polarization and good cycling stability in symmetric cells.

To further reveal the dendrite suppression effect in the quaternary melt, we first conducted morphological studies on Al plating in the two electrolytes at the same current density of 1 mA cm⁻² with an areal capacity of 3 mAh cm⁻². In the ionic liquid electrolyte, the Al deposits show many tiny particles with size below 2 μm and large aggregates (Supplementary Fig. 4a). The corresponding EDS elemental mapping image confirms the existence of only Al deposits on the Mo substrate (Supplementary Fig. 4b). In great contrast, the Al deposits in the quaternary melt electrolyte exhibit compact micron-sized crystals with well-defined facts and relatively large size above 20 μm (Supplementary Fig. 4c, d), which are expected to suppress dendrites-inducded short circuit during cycling. On the basis of the typical heterogeneous nuclei theory[34], the obvious difference in the two electrolytes is attributed to lower surface tension between nucleus and electrolyte, and faster diffusion kinetics in the quaternary melt. Furthermore, time-of-flight secondary ion mass spectrometry (TOF-SIMS) was performed to identify the chemical compositions near the surface of the Al anode after cycling in the ionic liquid and quaternary alkali melt electrolytes. Importantly, the depth distribution was investigated by using ion beam to remove layers from the anode surface for analysis (Supplementary Fig. 5). For the ionic liquid electrolyte, the $C^+$, $AlCl^+$ and $CH_2^+$ secondary ions show a large presence on the surface, and an obvious decrease along with the increase of depth. For example, a 3D depth rendering image of $C^+$ confirms that the carbon component is enriched in the outer surface. This indicates the formation of the solid electrolyte interphase (SEI) on the Al surface with the ionic liquid electrolyte, which consists of a mixture of organic and inorganic components. In great contrast, with the quaternary alkali melt electrolyte, the intensity of $C^+$, $AlCl^+$ and $Cl^+$ is about two orders of magnitude lower than that observed for the ionic liquids; and no $CH_2^+$ is observed. This indicates that as none organic components are present, the alkali melt does not undergo reactions with aluminum metal and no SEI is observed. This difference in the interphase between the two electrolytes is confirmed by 2D chemical ion images (Supplementary Figs. 6 and 7). Generally, the absence of SEI components on the Al anode surface also leads to non-dendritic growth in the quaternary alkali chloroaluminate melt electrolyte.

## Electrochemical performances with an optimized cathode

With the above-mentioned fast Al³⁺ desovlation kinetics in the quaternary melt electrolyte, the intrinsic conversion reaction kinetics of sulfur is another limiting factor for the rate capability of an Al−S battery at a low temperature of 85 °C. Nanostructured metal oxides/sulfides have been widely used to promote sulfur reaction in Li−S

batteries and Al–S fitted with organic electrolytes[35], but we find that most of them are chemically unstable in the highly active alkali chloride melts (Supplementary Fig. 8). Instead, we designed a lightweight and chemically stable nitrogen-doped carbon framework (denoted as NCF) as the sulfur host. Structural characterizations of the NCF show the presence of rich N-induced defects with an interconnected hollow structure and a high surface area of 1642.9 m² g⁻¹ (Supplementary Figs. 9 and 10). A simple melting diffusion method was applied to infiltrate sulfur onto the NCF to obtain the sulfur/NCF composite (S/NCF) cathode. Structural, morphological, and chemical descriptions of the composites are discussed in detail (Supplementary Fig. 11).

To evaluate the electrochemical performance, molten salt Al−S cells were assembled using the S/NCF cathode, Al foil anode and the quaternary melt electrolyte. In addition, cells with the ionic liquid EMIC−AlCl₃ electrolyte or the plain carbon (Ketjen black)-based sulfur composite cathode (noted as S/KB) were assembled for comparison. The chosen voltage range depends on the redox potential of the active electrode materials under different reaction conditions. However, due to sluggish sulfur conversion reaction kinetics at room temperature (25 °C), we expanded the voltage window to 0.2−1.6 V to evaluate the electrochemical performance of the EMIC−AlCl₃-based Al−S cell. The EMIC−AlCl₃-based Al−S cell using S/NCF cathode shows a low discharge capacity of 262 mAh g⁻¹ and a large voltage polarization of -1.06 V at a rate of C/5 at 25 °C (Fig. 3a). When the operation temperature increases to 85 °C, the cell shows an increased discharge capacity of 374 mAh g⁻¹ and reduced polarization of -0.60 V, but this is accompanied by a very low Coulombic efficiency (CE) of 23%. This means although increasing the temperature improves the overall reaction kinetics, the high solubility of aluminum−polysulfides (if present) in EMIC−AlCl₃ leads to serious shuttle effect. In great contrast, at the same conditions (85 °C and C/5), the Al−S cell fitted with quaternary alkali melt displays a high capacity of 931 mAh g⁻¹ and a lower voltage polarization of -0.37 V (Fig. 3a). This difference is attributed to higher desolvation and reaction kinetics and yet lower polysulfide solubility in the quaternary melt. The highly reproducible cyclic voltammetry curves over cycling further evidences a highly reversible conversion reaction (Supplementary Fig. 12a). The galvanostatic intermittent titration technique (GITT) profile shows a quasi-equilibrium cell voltage of -1.03 V, wherein, the discharge overpotential is higher than that of the charge (0.19 vs. 0.07 V), indicating an asymmetric conversion reaction mechanism over discharging and charging (Supplementary Fig. 12b), which will be discussed in the next section. Fortunately, for such batteries to be used for mobile or static energy storage, it is often more important to exhibit fast-charging capability than fast-discharging capability.

Due to the above-mentioned asymmetric reactions, we use different charge and discharge rates to evaluate the rate and cycling performance. For clarity, the notation of 1D (and 1C, a current density of 1675 mA g⁻¹) is used to denote a discharge (and charge) rate for full discharge (and charge) within one hour. With a charging rate of C/2, the S/NCF cathode in the quaternary melt shows a capacity of 725 mAh g⁻¹ with a low voltage polarization of -0.34 V and a high CE of 93% at a discharging rate of D/5, which is superior than that of the S/KB cathode (capacity: 466 mAh g⁻¹, polarization: 0.46 V; Fig. 3b, c). Notably, the cell shows little capacity fading and very high voltage stability over 100 cycles, superior to the S/KB cathode (Fig. 3c, and Supplementary Fig. 12c). When tested at a lower charging rate of C/5, the cell using S/NCF cathode retains a high discharge capacity of 869 mAh g⁻¹ (with high capacity retention of 93.4%) over 200 cycles, superior than that using S/KB cathode (retained capacity of 742 mAh g⁻¹ over 140 cycles, Fig. 3d and Supplementary Fig. 12d). This much reduced polarization and enhanced cycling stability is obviously attributed to the nitrogen-doped carbon that facilitates the sulfur conversion reaction. The polysulfide shuttle phenomenon is still present in the molten

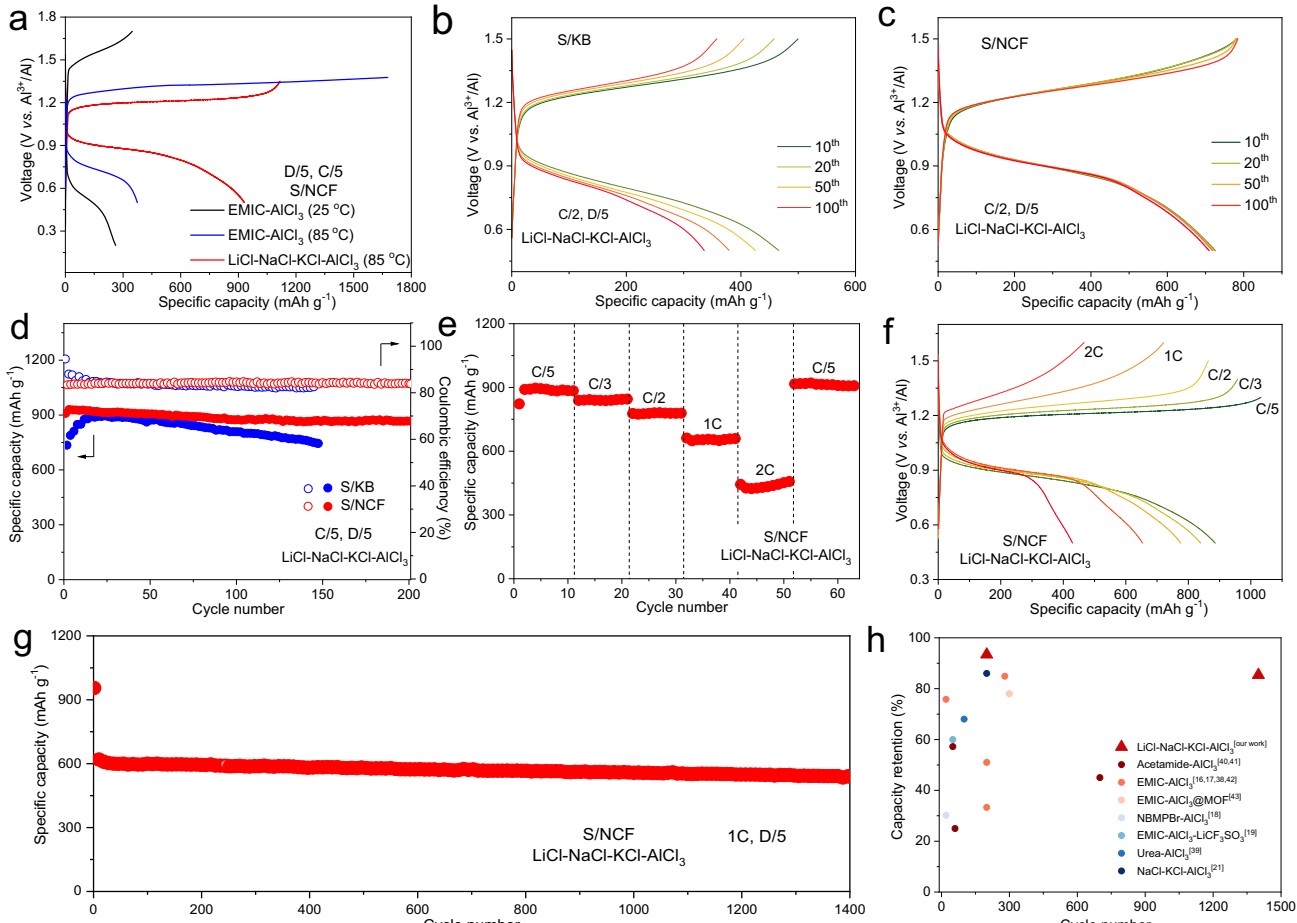

**Fig. 3 | Electrochemical performances of the Al–S cells at 85 °C. a** The voltage profiles of the Al–S cells with a S/CNF cathode, using the quaternary melt and ionic liquid electrolytes at different operation temperatures. **b**, c The voltage profiles of the Al–S cells using quaternary melt, and a S/KB cathode (**b**) or a S/CNF cathode (**c**) with a discharging rate of D/5 and a charging rate of C/2 at 85 °C. **d** Cycling performances of the S/NCF and S/KB composite cathodes at discharging/charging rate of C/5 and 85 °C. **e** The rate performance and **f** the corresponding voltage profiles of the S/NCF composite at a fixed discharging rate of D/5 and varied charging rates ranging from C/5 to 2 C at 85 °C. **g** Long-term cycling performance of the S/NCF composite cathode at a discharging rate of D/5 and a high charging rate of 1 C at 85 °C. **h** Comparison of the electrochemical performances demonstrated in our work and previous reports, in terms of the capacity retention and the cycle numbers; the reference numbers are labeled.

salt, but much less than that in the ionic liquid (Fig. 3a); but importantly, it does not lead to capacity fading as in the typical case for Li–S batteries. This clearly indicates that the partially dissolved aluminum polysulfides do not form inactive solids on Al surface over the course of shuttling, which is evidenced by the sulfur-free Al anode in molten salt Al–S battery after cycling (Supplementary Fig. 13). This is critically distinguished from conventionally perceived sulfur electrochemistry and we propose that it is foundational for the high cycling stability of molten salt Al–S batteries. Further efforts in tuning the chloroaluminate melt chemistry, designing high-surface-area/catalytic hosts and modifying separators are expected to allow further improvement on the CE[36,37]. The electrode with a sulfur loading of 3.1 mg cm⁻² exhibits a high reversible capacity of 760 mAh g⁻¹ and good cycling stability at 85 °C and C/5 (Supplementary Fig. 14a, b). Even at a high sulfur loading of 6.5 mg cm⁻² and C/10, the electrode shows an obvious capacity increase up to 530 mAh g⁻¹ after 40 cycles (Supplementary Fig. 14c, d). To achieve higher cycling performance of Al–S battery at a higher mass loading, further efforts on cathode design and electrolyte formulations are required.

Further, the charging rate capability was evaluated at different charging rates from C/5 to 2C at a fixed discharging rate of D/5. Using the quaternary alkali chloroaluminate melt and S/NCF cathode, the Al–S cell exhibits discharge capacities of 895, 851, 779, 665 and

426 mAh g⁻¹ at C/5, C/3, C/2, 1C, and 2C, respectively (Fig. 3e). When the rate returns to C/5, the capacity recovers to 913 mAh g⁻¹, indicating excellent high-rate stability. The corresponding voltage profiles at different charging rates show a single-plateau behavior that features faradaic reactions (Fig. 3f). Impressively, at a charging rate of 1 C, the cell delivers a high discharge capacity of 542 mAh g⁻¹ and a high capacity retention of 85.4% over 1400 cycles (Fig. 3g). Compared with the state-of-the-art Al–S batteries in the literature, the molten salt Al–S battery operated at 85 °C possesses excellent rate capability and long-term cycling stability (Fig. 3h, detailed in Supplementary Table 2)[16–19,21,38–43].

**Revealing the asymmetric reaction mechanism**

The sulfur reaction mechanism in the quaternary alkali chloroaluminate melt electrolyte was studied to seek the origin of the high sulfur conversion kinetics. We first thoroughly studied the sulfur cathodes at full discharge. The major X-ray diffraction peaks are well indexed to Al₂S₃ (JCPDS No. 47-1314), confirming a reaction predominantly forming Al₂S₃ (Fig. 4a). Further, the transmission electron microscopy images show the formation of abundant crystalline nanoparticles with average size of ~2 nm (Fig. 4b and Supplementary Fig. 15a–d), which is confirmed to be Al₂S₃ by lattice indexing. Specifically, the ordered lattice fringes with spacing

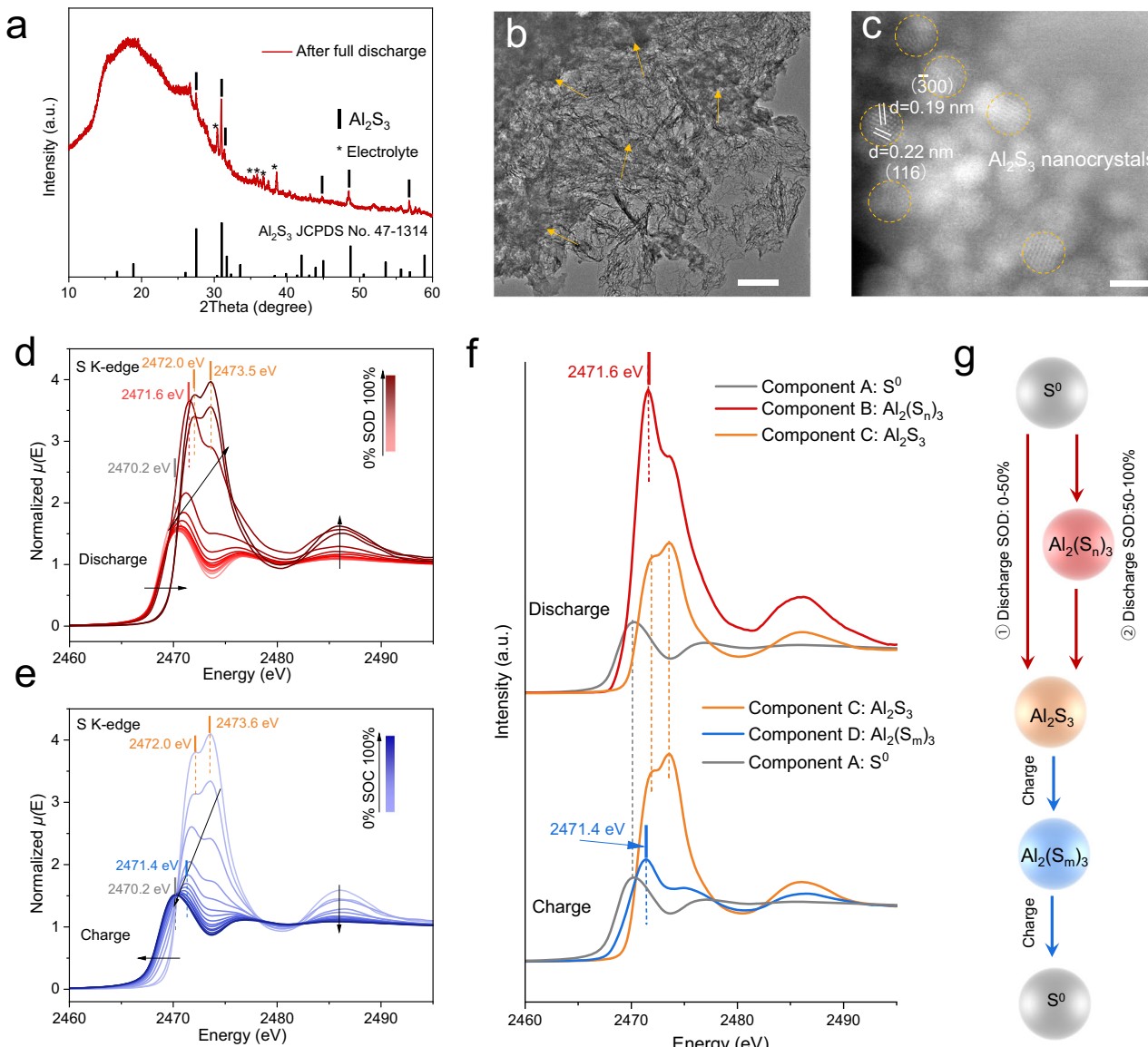

**Fig. 4 | Understandings on the reaction mechanism of the Al–S battery using quaternary alkali chloroaluminate melt electrolyte during discharge and charge at 85 °C. a** XRD pattern of the sulfur cathode after full discharge. The diffraction peaks marked by asterisks belong to the residual chloroaluminate melt electrolyte. **b, c** TEM images of the sulfur cathode after full discharge, **b** scale bar: 200 nm; **c** scale bar: 2 nm. **d, e** The ensembles of S K-edge XANES spectra of the sulfur cathode during the discharge (**d**) and charge (**e**) of the Al–S cell. **f** S K-edge XANES spectra of the three identified primary components during discharge, including $S^0$ (A, black), $Al_2(S_n)_3$ (B, red), and $Al_2S_3$ (C, orange), and the three during charge, including $S^0$ (A, black), $Al_2(S_m)_3$ (D, bule) and $Al_2S_3$ (C, orange). **g** Schematic illustration of the proposed reaction pathways of the molten salt Al–S battery; during discharge, the sulfur experiences a two-phase conversion reaction in the SOD of 0–50% and a three-phase conversion process in the SOD of 50–100%, finally forming $Al_2S_3$ product; during charge, the $Al_2S_3$ converts into the $S^0$ through a three-phase conversion process.

distances of about 0.19 and 0.22 nm correspond to the ($\bar{3}$00) and (116) planes of $Al_2S_3$ (Fig. 4c), and the [0$\bar{6}$1] zone axis for $Al_2S_3$ phase is confirmed by the fast Fourier transformation (FFT) pattern (Supplementary Fig. 15e, f). These evidences combined firmly prove $Al_2S_3$ as the terminal discharge product. Further, the S 2$p$ XPS spectrum reveals the chemical states of the discharged sulfur cathode. With the pristine sulfur cathode as the reference that shows two doublet peaks at 165.3 and 164.1 eV (S 2$p_{1/2}$ and 2$p_{3/2}$ states of $S_8$), the spectrum of the S/NCF cathode after full discharge clearly covers a wide range of binding energy and can only be fitted with at least six doublet peaks, which are indexed to $S_8$, polysulfides ($S_n^{2-}$) and $Al_2S_3$ (Supplementary Fig. 15g)[41,44]. This indicates the presence of polysulfide intermediates, which in this case is attributed to incompletely reacted sulfur.

The single-plateau behavior of molten salt Al–S battery here is much distinguished from the two-plateau behaviors of conventional Li–S/Na–S batteries. To reveal the sulfur reaction pathway during cycling, local structural changes of sulfur during the first cycle were studied by *operando* X-ray absorption near-edge spectroscopy (XANES). The sulfur K-edge absorption occurs due to an electron transition from 1$s$ states to unoccupied 3$p$ states[45]. The pristine sulfur cathode shows a feature peak located at 2470.5 eV, owed to the S 1$s$ to S–S $\pi^*$ state transition ("white line") of elemental sulfur[46]. First, the XANES spectra of the sulfur cathode show a rather reversible evolution over the first cycle, indicating a highly reversible sulfur redox reaction process (Supplementary Fig. 16a). During discharge, the white line intensity gradually increases owed to the population of electron in the

valence 3p orbitals indicating reduction of sulfur, while another peak at 2471.6 eV shows up which indicates the presence of intermediate(s) (Fig. 4d)[46,47]. At the end of discharge, two signature peaks at 2472.2 and 2473.5 eV appear, which we tentatively ascribe to $Al_2S_3$ present in the reactive melt[48]. The increased binding energy for $Al_2S_3$ is due to the S 1s to σ* transition and the strong electron binding ability in $Al_2S_3$[49]. For the charging process, the evolution of XANES spectra shows the reversed trend (sulfur oxidation), but the dynamics and exact species of intermediates are different as discussed below (Fig. 4e).

Notably, the ensemble of XANES spectra on discharge shows absence of isosbestic points (i.e. unique common intersection points), suggesting more than two components are present over the whole discharge (Fig. 4d, and Supplementary Fig. 16b, c). Also, the principal component analysis statistically confirms that a minimum of three components is required to describe the ensemble of spectra (Supplementary Fig. 17). Using the multivariate curve resolution-alternating least-squares (MCR-ALS) method[50,51], we de-convolved the ensemble of spectra on discharge and identified the three probable primary components that mathematically describes all spectra, which we ascribe to $S^0$, $Al_2(S_n)_3$, and $Al_2S_3$ (Fig. 4f). Deconvolution of the ensemble of spectra on charge similarly yields three components, among which the two terminal products of $S^0$ and $Al_2S_3$ are common with the discharge spectra, while with a different intermediate that features a different absorption intensity at 2471.6 eV (Fig. 4f). We here tentatively assign it as $Al_2(S_m)_3$ with a different chain length than that of $Al_2(S_n)_3$[52]. This phenomenon indicates the asymmetric reaction mechanism of molten salt Al-S battery over discharge/charge. We note that due to the presence of sulfur self-absorption effect here, the signal-to-noise ratio in the fluorescence-mode spectra disallows any local structural fitting of the extended X-ray absorption fine spectra (EXAFS). Future efforts to confirm the precise structures and chain-length of aluminum polysulfides are ongoing.

Further, linear combination fitting analysis of the XANES spectra using the three identified components as standards was performed to reveal the dynamics of sulfur reactions (Supplementary Fig. 16d, e). At a state of discharge (SOD) of 0–50%, the $S^0$ directly converts to $Al_2S_3$ without forming $Al_2(S_n)_3$, indicating a two-phase reaction process (proposed Eq. (1)). Note that at this stage, the conversion dynamics of $S^0$ seems abnormally low, the error of which is possibly due to the dynamic decrease of sulfur particle size along discharge and thus varied degree of sulfur self-adsorption effect. Upon further discharge, $Al_2(S_n)_3$ forms, implying that the residual $S^0$ may undergo a three-phase stepwise conversion ($S^0$–$Al_2(S_n)_3$–$Al_2S_3$) (proposed Eqs. (2) and (3)). As proposed in Fig. 4g, the $S^0$ undergoes two reaction pathways to form the end $Al_2S_3$ product at different stages of discharge. On charge, the $Al_2S_3$ undergoes a multi-phase stepwise conversion over the whole charge process ($Al_2S_3$–$Al_2(S_m)_3$–$S^0$) (Eqs. (4) and (5); Fig. 4g), which is different from the discharge. We propose that the formation of the $Al_2(S_m)_3$ intermediate over the whole charge is crucial for rapid-charging for Al-S battery at 85 °C. Note that the $Al^{3+}$ desolvation from the chloroaluminate electrolyte occurs prior to any reaction (proposed equation 0). After all, we note that the analyses above should be considered as semi-quantitative owed to the presence of appreciable sulfur self-absorption effect; but nevertheless, the semi-quantifications provide coherent implications on the asymmetric sulfur reactions featuring fast charging. Our *operando* electrochemical impedance spectroscopy study that monitors the evolution of interfacial charge transfer provides further evidence (discussed in detail in Supplementary Fig. 18).

The $Al^{3+}$ desolvation:

$$4Al_nCl_{3n+1}^- \rightarrow (n-1)Al^{3+} + (3n+1)AlCl_4^- \tag{0}$$

Reactions on discharge:

$$3S^0 + 2Al^{3+} + 6e^- \rightarrow Al_2S_3 \tag{1}$$

$$3nS^0 + 2Al^{3+} + 6e^- \rightarrow Al_2(S_n)_3 \tag{2}$$

$$Al_2(S_n)_3 + 2(n-1)Al^{3+} + 6(n-1)e^- \rightarrow nAl_2S_3 \tag{3}$$

Reactions on charge:

$$mAl_2S_3 - 6(m-1)e^- \rightarrow Al_2(S_m)_3 + 2(m-1)Al^{3+} \tag{4}$$

$$Al_2(S_m)_3 - 6e^- \rightarrow 3mS^0 + 2Al^{3+} \tag{5}$$

## Discussion

In summary, we have demonstrated a resourcefully sustainable rechargeable Al–S battery operated at 85 °C enabled by a quaternary alkali chloroaluminate melt electrolyte, which shows rapid-charging capability and long-term cycling stability. Based on experimental results and simulation analyses, the solvation structure of the quaternary melt electrolyte shows the presence of high-order $Al_3Cl_{10}^-$ and $Al_4Cl_{13}^-$ as well as $Al_2Cl_6$ clusters, which is crucial for low $Al^{3+}$ desolvation barrier and fast reaction kinetics. Moreover, the nitrogen-doped carbon host is chemically compatible with the electrolyte and facilitates the sulfur reaction yielding higher specific capacity and low polarization. Benefitting from the merits of both electrolyte and electrode, the Al–S battery shows a discharge capacity of 542 mAh g⁻¹ with high capacity retention of 85.4% over 1400 cycles at a rate of 1 C at 85 °C, demonstrating excellent cycling stability. Importantly, our comprehensive characterizations reveal a reversible and yet asymmetric multi-phase sulfur reaction mechanism involving aluminum polysulfides, accounting for the rapid-charging capability at 85 °C. Our work shows that it is possible to construct molten salt batteries that can operate at a very low temperature, counterintuitive to previous reports that grant molten salt electrochemical systems with a sign of overly high temperature. We believe this work will encourage more future work in exploring molten salt electrolytes with even lower temperature operation to realize fully practical, low-cost, highly safe and fast-charging aluminum batteries.

## Methods

### Electrolyte preparation

All the operations below were performed inside the argon-filled glovebox (condition: $O_2 < 0.01$ ppm, $H_2O < 0.01$ ppm). For the preparation of molten salt electrolytes, anhydrous $AlCl_3$ (Acros, 99.9%), NaCl (Aladdin, 99.9%), LiCl (Aladdin, 99.9%), and KCl (Aladdin, 99.9%) with an optimized molar ratio of 1.2:0.43:0.42:0.15 were added into a glass weighing flask. The flask was sealed and heated to 120 °C in an oven for 12 h, yielding a homogeneous clear liquid. After cooling down to room temperature, the solid mixture was grinded thoroughly to obtain the powder electrolyte. For the preparation of the organic ionic liquid electrolyte, anhydrous $AlCl_3$ and 1-ethyl-3-methylimidazolium chloride (EMIC, Sigma-Aldrich, 98%) with a mole ratio of 1.2:1 were mixed thoroughly. The EMIC was dried at 150 °C for 12 h before use in order to remove the residual water via a melting and recrystallization process. During the mixing process, the anhydrous $AlCl_3$ was slowly added into EMIC at room temperature under rigorous stirring, eventually forming a transparent liquid.

### Synthesis of sulfur/carbon composites

All the chemicals were of reagent-grade and used without further purification. The commercial ZnO nanoparticles (Macklin, 99.9%) and

2-methylimidazole (Macklin, 98%) were separately contained and treated at 140 °C for 6 h under a low-pressure in a vacuum oven. After low-pressure vapor super-assembly, a thin zeolitic imidazolate framework (ZIF-8) coating was formed on the surface of ZnO nanoparticles. Subsequently, the ZnO@ZIF-8 nanoparticles were calcined in $N_2$ at 650 °C for 3 h, forming the ZnO@NCF nanoparticles. Then, the ZnO@NCF particles were immersed into hydrochloric acid solution (5 wt%) under ultrasonication to remove ZnO. Subsequently, after treatment at 1000 °C for 3 h, the NCF material was obtained. The sulfur composites with 60 wt% of sulfur and 40 wt% of NCF (or Ketjen Black carbon) were prepared by a melting diffusion method at 155 °C for 12 h.

### Theoretical calculations

The calculations were performed by using a first-principles method based on density functional theory (DFT) with the generalized gradient approximation (GGA) in the form of Perdew–Burke–Ernzerhof (PBE) exchange-correlation functional[53], as implemented in the Vienna Ab initio Simulation Package (VASP 6.3.2)[54]. Long-range van der Waals dispersion interactions were treated using the DFT-D3 method of Grimme[55]. The wave functions are expanded in plane waves up to a kinetic energy cutoff of 520 eV. The Brillouin zone (BZ) integrals were performed using a Monkhorst–Pack sampling scheme at the Γ point only[56]. The unit cell lattice parameters (unit cell shape and size) and atomic coordinates were fully relaxed in each system until the forces on all the atoms were smaller than 0.01 eV Å$^{-1}$. We simulated the bulk electrolytes using a computational supercell (≥14 Å in lattice constant) that consisted of about 200 atoms, with periodic boundary conditions along all directions. The $AlCl_3$–(Li/Na/K)Cl electrolyte system was simulated with molar ratios 1.2:1, while the $AlCl_3$-EMIC system also features a molar ratio of 1.2:1. In all simulations, the density of the electrolytes were fixed close to the experimentally measured values. The initial configuration was prepared by avoiding adjacent ions with the same charge; the number of each type of ions or molecules was determined by the molar ratio and experimentally measured density. Each system was equilibrated at 358 K in the canonical ensemble (NVT)[57] for 10 ps with a time step of 1 fs. A constant temperature condition was maintained using a Nose–Hoover thermostat.

### Materials characterizations

The XRD characterization was measured using a D8 Advance X-ray diffractometer with a Cu Kα X-ray source. The Raman spectra were collected using a Renishaw INVIA micro-Raman spectroscopy system. The XPS measurement was performed using a VG MultiLab 2000 instrument. The SEM images were obtained using a JEOL JSM-7100F at an acceleration voltage of 20 kV. Elemental mapping was conducted by using an EDX-GENESIS 60S spectrometer. The Brunauer–Emmett–Teller (BET) and Barret–Joyner–Halenda (BJH) plots was performed from nitrogen adsorption isotherms collected at 77 K using a Tristar-3020 instrument. Transmission electron microscopy (TEM), scanning transmission electron microscopy (STEM), and energy dispersive X-ray spectroscopy (EDS) mapping images were taken by ThermoFisher Titan Themis G2 60-300. To explore the discharged product of the S/NCF electrode, the discharged cells were disassembled in the glovebox, and the corresponding S/NCF cathode was used for further characterizations. To investigate the morphology and structure of the Al anode surface in Al–Mo and Al–S cells, the resulting Al anode after cell disassembly was fully washed by hexane to remove the residual electrolytes. Time-of-flight-secondary ion mass spectrometry (TOF-SIMS) measurement was carried out using a PHI nanoTOF 3 to analyze the depth profiles of different secondary ions on the Al anode surface.

### Electrochemical characterizations

The cathode slurry was prepared by mixing 80 wt% S/NCF or S/KB, 10 wt% Super P and 10 wt% polytetrafluoroethylene (PTFE) in isopropanol. After stirring for 12 h, the slurry was dropped onto the carbon paper. The areal loading of sulfur per unit area was around 1.50 mg cm$^2$. For the Al–S battery, a house-designed Swagelok® cell with a protection sheath was assembled using a commercial Al foil as the anode, a glass fiber as the separator, the quaternary chloroaluminate melt as the electrolyte and S/KB (S/NCF) as the cathode. The thickness of commercial Al foils is 0.2 mm. The dosage of electrolyte in each cell is ~100 mg. These cells were placed into a constant temperature chamber at 85 °C. For the exchange current test, the linear scanning voltammetry (LSV) in Al|Al symmetric cell was performed at a scan rate of 1 mV s$^{-1}$ with the voltage range limited to −35 mV from open-circuit voltage (OCV). The ionic conductivity of the quaternary alkali chloroaluminate melt electrolyte was investigated using a Mo–Mo configuration. At 85 °C, the quaternary melt was immersed into the whole glass fiber separator. The ionic conductivity of the electrolytes is usually calculated using electrochemical impedance spectroscopy by analyzing the Nyquist plot. Generally, the ohmic resistance ($R_b$) is the intercept at the real axis in the Nyquist plot. Therefore, the ionic conductivity ($σ$) is calculated by the equation: $σ = d/(R_b*A)$; where $d$ is the distance of Mo–Mo electrodes, and $A$ is the geometric area of the electrolyte electrode. Galvanostatic charge–discharge and the galvanostatic intermittent titration technique (GITT) measurements were performed using a multichannel battery testing system (NEWARE). The CVs, LSV curves and in situ EIS spectra were collected using a Bio-logic SP-200. Each EIS was measured in a frequency range from 20k Hz to 0.02 Hz with 91 points. The in situ EIS measurement of the molten salt Al–S battery was conducted repetitively after a fixed capacity interval of 100 mAh g$^{-1}$.

### In situ XANES measurement and analysis

The S K-edge XANES experiments were carried out at the sector 9-BM-B in the Advanced Photon Source, the Argonne National Laboratory. The XANES data were collected in fluorescence mode using a four-element vortex detector with a Si(111) crystal monochromator under helium flow. The calibration of binding energy was carried out using sodium thiosulfate pentahydrate. The *operando* XANES measurements were performed on 2032 coin cells that use a Kapton window on the cathode side, which was similar with our previous report. In detail, a glassy carbon disk was placed between the Kapton window and the positive electrode, which not only serves as a current collector but also keeps uniform pressure in the whole cell. A sulfur cathode, an Al foil anode, a glass fiber separator and the molten salt electrolyte were used for the *operando* cell. To avoid the cell from electrolyte corrosion, Mo foil was placed on the anode side of the cell. The Athena software was utilized for calibration, alignment, and normalization of the XANES spectra. Principal component analysis (PCA) in the energy space was employed to identify the number of main components contributing to the ensemble of XANES spectra. The multivariate curve resolution-alternating least-squares (MCR-ALS) method was used to deconvolute the ensemble of spectra. By performing linear combination fitting of the spectra using the main components as standards, the fractions of each component were quantified.

## Data availability

The data that support the plots within this paper and other findings of this study are available from the corresponding author upon reasonable request. Source data are provided with this paper.

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

## Acknowledgements

This work was supported by the National Natural Science Foundation of China (NFSC, 92372115). We also acknowledge the support from National Key Research and Development Program of China (grant no. 2022YFE0198600); the National Natural Science Foundation of China (NFSC, grant nos. 22075002, and 52103329), and China National Petroleum Corporation-Peking University Strategic Cooperation Project of Fundamental Research. This research used resources of the Advanced Photon Source, an Office of Science User Facility operated for the US Department of Energy (DOE) Office of Science by Argonne National Laboratory and was supported by the US DOE under Contract No. DE-AC02-06CH11357.

## Author contributions

Q.P., L.M., and J.M. conceived the concept. Q.P. and J.M. designed the experimental work. J.M., X.H., and Z.X. prepared the electrolytes and performed the physical characterization of the electrolytes and electrodes. J.M., Z.X., and Y.J. performed electrochemical performance measurements. L.X. and L.Z. performed theoretical calculations. F.L. and Y.J. participated in the XRD and XPS studies, as well as electron microscopy characterizations. Q.P., L.M., and J.M. performed the X-ray absorption near-edge spectroscopy analysis and proposed the reaction mechanism. All authors have thoroughly discussed the analysis of the data. Q.P., L.M., and J.M. wrote the manuscripts with contributions from all authors. Q.P. supervised the work.

## Competing interests

The authors declare no competing interests.
