## [Peer Review File · Nature Communications]

Rapid-charging aluminum-sulfur batteries operated at 85 °C with a quaternary molten salt electrolyteREVIEWER COMMENTS

Reviewer #1 (Remarks to the Author):

This manuscript presents a study on a quaternary molten salt electrolyte (including AlCl_3 , LiCl , KCl , and NaCl) for application in Al-S batteries. The authors emphasize the noteworthy operational aspect of these batteries, characterized by a relatively low temperature of operation at 85°C , as well as their favorable cyclic stability. This work is comprehensive, incorporating a substantial amount of experimental data encompassing battery performance and synchrotron-based X-ray absorption spectra. However, the major weakness of this work is the absence of a clear exposition regarding the significance of the proposed quaternary molten salt electrolyte in relation to previously reported binary and ternary electrolyte systems. Overall, the reviewer recommends revision before publication. Some comments are listed below:

1. Binary or ternary molten salt electrolytes are widely reported for Al-metal batteries. The authors are suggested to undertake a comprehensive comparative analysis between the proposed quaternary electrolytes and the aforementioned binary/ternary counterparts.
2. The intrinsic relationship between the molar or mass ratios of AlCl_3 , KCl , LiCl , and NaCl holds substantial implications for the resultant battery performance. In light of this, it is recommended that the authors undertake a comprehensive examination of battery performance by systematically varying the aforementioned salt ratios.
3. The respective role of each AlCl_3 , KCl , LiCl and NaCl salt component should be further discussed.
4. The peaks in synchrotron soft X-ray data (Figure 4f) should be divided and fitted. Some quantitative information should be provided.
5. In Figure 3d, the Coulombic efficiency of battery is still very low. The authors should make more specific discussion. Some ongoing strategies to further improve the Coulombic efficiency should be discussed.
6. The electrolyte and electrode design are important for metal-S batteries. Some relevant work published recently may be referred such as Nature Communications 2021, 12, 7195; Nature Communications 2021, 12, 5714.

Reviewer #2 (Remarks to the Author):

In this work, Meng et al. report a rapid-charging Al-S battery operated at a sub-water-boiling temperature of 85°C with a tamed quaternary molten salt electrolyte. The manuscript lacks novelty, the analysis of the results is not profound enough, and the authors do not offer any constructive insights. This manuscript is not suitable for publication in Nature Communications. Some specific comments are below.

1. Some similar publications related to the concept of alkali chloroaluminate melt have been reported for Al-S batteries (e.g., Nature Comm. 2021, 12(1): 5714, Energy Environ. Sci., 2022, 15, 5229-5239). What are the differences and innovations in the current manuscript compared to other published articles like these?
2. The authors claimed that even though the dissolution of aluminum polysulfides is detected, it does not lead to capacity fading since the polysulfide species do not form inactive solids on Al surface at the anode over the course of shuttling. However, this claim lacks supporting characterization of the Al-anode. Furthermore, the long-term cycling

performance at C/5 and 1C reveals a low coulombic efficiency of 85%. Do the authors provide an explanation for the root cause of this low coulombic efficiency?

3. Ionic conductivity is one of the important parameters to evaluate the motion of ionic charge in the electrolyte. Have the authors tried measuring the conductivity of the alkali chloroaluminate melt electrolytes?

4. The authors mentioned that there is a dendrite suppression effect in the quaternary melt electrolyte based on the Al|Al symmetric cell results. Is there any proposed mechanism and experimental result to support this?

5. What is the function of KCl?

6. The authors claim that the energy density of Al-S batteries is moderate; however, the areal loading of sulfur is only 1.5 mg cm⁻². To explore the practical feasibility further, has the molten salt electrolyte system been tested with high sulfur loading in the range of 3 to 5 mg cm⁻²?

Reviewer #3 (Remarks to the Author):

This manuscript reports the utilization of a quaternary molten salt electrolyte for Al-S battery, which exhibit desirable high reversible capacity as well as outstanding cycling stability. The structure of the molten electrolyte as well as the charge/discharge working mechanism of the Al-S battery has been investigated with rational characterizations. The manuscript has been carried out with logical writing and well-supported conclusions. Therefore, I recommend the manuscript to be considered for publication after minor revisions as follows:

1. The melting points of the individual salts, as well as the binary and ternary salts should be supplemented for reference.
2. Previous reports have demonstrated the utilization of alkali metal chlorides as additives for ionic liquid electrolyte to modify the interface stability of Al anodes. Does the quaternary salt have similar functions? The compositions of the interfaces on the Al anodes should be investigated.
3. The authors have published results for Al-chalcogen batteries with the ternary NaCl-KCl-AlCl₃ salt in Nature 608, 704–711 (2022), therefore the necessity of the LiCl in the quaternary salt should be discussed.
4. The thickness of the Al foil and the dosage of electrolyte in each cell should be specified.

Reviewer #1 (Remarks to the Author):

This manuscript presents a study on a quaternary molten salt electrolyte (including AlCl_3 , LiCl , KCl , and NaCl) for application in Al-S batteries. The authors emphasize the noteworthy operational aspect of these batteries, characterized by a relatively low temperature of operation at 85°C , as well as their favorable cyclic stability. This work is comprehensive, incorporating a substantial amount of experimental data encompassing battery performance and synchrotron-based X-ray absorption spectra. However, the major weakness of this work is the absence of a clear exposition regarding the significance of the proposed quaternary molten salt electrolyte in relation to previously reported binary and ternary electrolyte systems. Overall, the reviewer recommends revision before publication. Some comments are listed below:

Our response: We fully appreciate the reviewer's thoughtful and encouraging comments on our manuscript, and offering the opportunity to address and clarify the issues raised in the report. We have now further analyzed and emphasized the significance of the proposed quaternary molten salt electrolyte in our revised manuscript. Our responses to the points raised in the report are described below following the reviewer's comments.

1. Binary or ternary molten salt electrolytes are widely reported for Al-metal batteries. The authors are suggested to undertake a comprehensive comparative analysis between the proposed quaternary electrolytes and the aforementioned binary/ternary counterparts.

Our response:

We appreciate the reviewer's valuable suggestion. When molten salts are used as electrolytes in batteries, the physical properties including melting point, viscosity and ionic conductivity should be considered. Therefore, we have made a comprehensive comparative analysis on the physical properties between our quaternary electrolytes and binary/ternary counterparts.

As one of the most important physical property of molten salts, we now first discuss the melting point, which directly determines the battery operation temperature. Molten salts are ionic mixtures that are solid at standard temperature and pressure but are liquid at elevated temperatures. For molten salts, the latent heat of fusion (ΔH_f) dictates the energy required to overcome the electrostatic attraction forces between cations and anions; a weaker electrostatic attraction force leads to a lower fusion temperature. The magnitude of ΔH_f highly depends upon the intermolecular forces involved in crystal binding and evidently varies for different types of salts. In addition, the variation of the fusion entropy (ΔS) plays great role. Because the Gibbs free energy of a specific molten salt system is constant during fusion, the fusion temperature is determined by the basic

relation of thermodynamics: $T_f = \Delta H_f / \Delta S$. On basis of the above equation, the increase of the entropy of the system contributes to decrease of the fusion temperature.

Specifically, it is understood that there are five principal mechanisms that can increase the system entropy of molten salts as follows: (a) increase in vibrational entropy; (b) increase of orientational randomization; (c) increase in positional disorder; (d) randomization of the internal configuration of molecules or ions containing groups; (e) changes of association or chemical bonding on melting (J. Chem. Educ. 1962, 39, 59). From the perspective of thermodynamics, in our quaternary melt, the presence of multiple components greatly increases the fusion entropy, leading to decreased fusion temperature. We now show the phase diagrams of some typical binary and ternary chloride molten salt systems (shown in Table R1 for the information of the referee; regenerated from the FactSage software). It is clear that our quaternary molten salt shows a much lower melting point ($\sim 80^\circ\text{C}$) than binary and ternary components.

Figure R1. Typical phase diagrams of binary and ternary molten salt systems.

Table R1. Summary on the melting point of chloride salts with varied ratios

Molten salts	Salt components	Molar ratios	Melting point ($^\circ\text{C}$)
Single	LiCl	1	605
	NaCl	1	801
	KCl	1	858
	AlCl_3	1	190

Binary	LiCl/NaCl	72/28	549
	KCl/NaCl	50/50	652
	KCl/LiCl	42/58	354
	AlCl ₃ /LiCl	58/42	110
	AlCl ₃ /NaCl	61/39	102
	AlCl ₃ /KCl	66/34	128
Ternary	AlCl ₃ /NaCl/KCl	61/26/13	~100
	AlCl ₃ /LiCl/KCl	59/29/12	~95
Quaternary	AlCl ₃ /LiCl/NaCl/KCl	120/42/43/15	~80

Second, the viscosity is another important physical property to evaluate the molten salts, which is highly related with the transport kinetics. The magnitude of viscosity for molten salts is determined by the operation temperature and structural units. The relationship between viscosity (η) and temperature (T) is described as follows: $\eta = Ae^{-E_{\eta}/RT}$. The E_{η} is the energy of activation for viscous flow. For molten salts, the electrical neutrality requires movement of both ions during viscous flow and the rate is determined by the larger ions. In general, highly ion-associated melts exhibit large values of E_{η} because the structural bonds must be broken before flow. In our previously reported work, the molten salt electrolytes for aluminium batteries contains catenated $Al_nCl_{3n+1}^-$ (e.g., $Al_2Cl_7^-$, $Al_3Cl_{10}^-$ and $Al_4Cl_{13}^-$) due to the presence of high levels of $AlCl_3$. The larger $Al_nCl_{3n+1}^-$ clusters lead to a higher viscosity. In alkali chloroaluminate melts, the alkali metal cations not only keep electric neutrality of molten salts, but also distribute around $Al_nCl_{3n+1}^-$ clusters, which are beneficial for lowering the E_{η} value. We propose that the presence of multiple alkali metal cations (e.g., Li^+ , Na^+ and K^+) in the quaternary molten salt can further break down the “net-work” structure owed to their asymmetric interactions. Therefore, compared with binary/ternary counterparts, our quaternary molten salt is projected to exhibit a lower E_{η} value, resulting in its lower viscosity at the same operation temperature. Unfortunately, it is very challenging at this moment to measure the exact viscosity of hot and air-sensitive liquids due to the lack of designed heating chamber; and therefore is subjected to further study.

Third, the ionic conductivity of molten salts is important and has to do with the nature of the conducting species, the structure and intermolecular forces in the melt, and the degree of dissociation. The ionic conductivity follows the temperature dependence as described in the Arrhenius equation: $k = A_k e^{-E_k/RT}$. From the perspective of thermodynamics, the activation energy for conductance (E_k) can be identified with the enthalpy of activation (ΔH_k) in theory. The entropy of activation (ΔS_k) is highly determined by the salt structures (as discussed above). Therefore, the increase of the system entropy in our quaternary molten salt can lead to the decrease

of E_k value and thus fast ionic conductivity. Further, the transport mechanisms in molten salts should be considered and discussed, which are highly dependent on the melt structures. For complex molten electrolytes containing polyatomic chain- or disc-like clusters, the transport mechanism involves a cooperative process, including simple ion jumping, rotation, migration or rapid reconfiguration of polyatomic clusters (J. Chem. Educ. 1962, 39, 59; J. Mol. Liq. 2017, 229, 330–338). The alkali metal ions during transport follow a simple ion jump mechanism due to their small size, while the catenated $Al_nCl_{3n+1}^-$ clusters experience rotation, migration or rapid reconfiguration process, which should be the rate-determining process for ionic conduction. Therefore, the key point to achieve high ionic conductivity of alkali chloroaluminate melts is to guarantee facile transport of catenated $Al_nCl_{3n+1}^-$ clusters in the applied field. In our quaternary molten salt, the presence of Li^+ , Na^+ and K^+ ions can further weaken the electrostatic force with catenated $Al_nCl_{3n+1}^-$ clusters because of the large size and more delocalized charge, and of the size variation induced asymmetric interactions, and thus facilitate the ionic conductivity.

In brief, compared with binary/ternary counterparts, our quaternary molten salt exhibits greater advantages in terms of melting point, viscosity, and ionic conductivity. **We have added these discussions to our revised manuscript.**

2. The intrinsic relationship between the molar or mass ratios of $AlCl_3$, KCl , $LiCl$, and $NaCl$ holds substantial implications for the resultant battery performance. In light of this, it is recommended that the authors undertake a comprehensive examination of battery performance by systematically varying the aforementioned salt ratios.

Our response:

We thank the reviewer's valuable suggestion and agree with the reviewer's comment.

The alkali chloroaluminate melts consist of $AlCl_3$ as Lewis acids and alkali metal chlorides as Lewis bases. By varying the ratio of $AlCl_3$ and alkali metal chlorides, different types of molten salts can be obtained. In our manuscript, our quaternary molten salt has a ratio of $AlCl_3$ and alkali metal chlorides ($LiCl$, $NaCl$ and KCl) with 1.2, which is considered as an acidic melt.

As suggested by the reviewer, two different molten salt electrolytes are prepared by varying the ratio of $AlCl_3$ and alkali metal chlorides, to understand the correlations between electrolyte ratios and battery performance. One is a basic melt with an $AlCl_3$ /alkali metal chloride ratio of 1.0, and the other is a more acidic melt with a ratio of 1.5; the ratios between the different alkali metal ions remains fixed. First, these ratios are off the eutectic ratio and therefore their melting point will be increased to some extent. Further, we have performed the electrochemical performance of Al-S batteries using the two electrolytes. As shown in Figure R2, the Al-S battery in the basic melt exhibits a low discharge capacity below 10 mAh g^{-1} and large voltage polarization at $85 \text{ }^\circ\text{C}$ and 0.2C, indicating sluggish reaction kinetics in basic melt. However, in the acidic melt with a ratio of 1.5, the resulting Al-S battery shows a high reversible capacity of $\sim 830 \text{ mAh g}^{-1}$ and a relatively low capacity retention of $\sim 74\%$ after 50 cycles. Therefore, indeed as the reviewer suggested, the salt ratios of the melt electrolytes have great impact in their physical properties (e.g., melting point, viscosity and ionic conductivity), and determines the battery electrochemical performance.

We have added these discussions in our revised manuscript.

Figure R2. Electrochemical performance of Al-S battery using different molten salt electrolytes at 85 °C and 0.2C. (a, b) Cycling performance and voltage profiles in the basic melt with an AlCl₃/alkali metal chloride ratio of 1. (c, d) Cycling performance and voltage profiles in the acidic melt with a ratio of 1.5.

3. The respective role of each AlCl₃, KCl, LiCl and NaCl salt component should be further discussed.

Our response:

We appreciate the reviewer's valuable suggestion. As discussed in the above question, the alkali chloroaluminate melts consist AlCl₃ as Lewis acids and alkali metal chlorides (KCl, LiCl and NaCl) as Lewis bases. Due to relatively high levels of AlCl₃, the AlCl₃ species participate in the formation of catenated Al_nCl_{3n+1}⁻ (e.g., Al₂Cl₇⁻, Al₃Cl₁₀⁻ and Al₄Cl₁₃⁻) in the alkali chloroaluminate melt, thus facilitating Al³⁺ desolvation kinetics. In the melt, the KCl, LiCl and NaCl components exist in ionic states (i.e., K⁺, Li⁺ and Na⁺), acting as the counter ions around the catenated Al_nCl_{3n+1}⁻ clusters. The coexistence of K⁺, Li⁺ and Na⁺ can further break down the "net-work" structure of the catenated Al_nCl_{3n+1}⁻ clusters by their asymmetric interactions, which is beneficial for low melting point, low viscosity and high ionic conductivity. It is noted that the ratio of KCl, LiCl and NaCl components is very important for the physical properties of the resulting melts. In our quaternary molten salt, the ratio of alkali metal chlorides has been optimized, thus leading to a low melting point.

4. The peaks in synchrotron soft X-ray data (Figure 4f) should be divided and fitted. Some quantitative information should be provided.

Our response:

We appreciate the reviewer's comment. Fig. 4f shows the S K-edge XANES spectra of the three identified primary components during discharge and charge. These primary components, which are identified based on mathematical de-convolution, the multivariate curve resolution-alternating least-squares method, can mathematically describe all spectra (Fig. 4d, e). Therefore, the S K-edge XANES spectra of identified primary components are already the basic spectra of single components, and therefore need not to be further divided. In fact, the identification of the primary spectra is somewhat equivalent to what the reviewer suggested, peak fitting.

Further, we have performed the linear combination fitting analysis of the XANES spectra using the three identified components as standards to reveal the dynamics of sulfur reactions. The detailed quantitative information is shown in Supplementary Fig. 19d, e. The asymmetric sulfur reaction processes are discussed in detail in our original manuscript.

5. In Figure 3d, the Coulombic efficiency of battery is still very low. The authors should make more specific discussion. Some ongoing strategies to further improve the Coulombic efficiency should be discussed.

Our response:

We appreciate the reviewer's valuable suggestion. We have made further discussions on the Coulombic efficiency of the proposed battery. As shown in Fig. 3d, the Al-S battery using quaternary alkali melt displays an average Coulombic efficiency of ~84%, which can be attributed to the dissolution of polysulfide intermediates in the melt electrolyte and thus polysulfide shuttle during charge. Importantly, it does not lead to capacity fading (Fig. 3d), which clearly indicates that the partially dissolved aluminium polysulfides do not form inactive solids on Al surface over the course of shuttling (as demonstrated in Figure R3). This phenomenon is different from the typical cases for Li-S batteries. At the same conditions (85 °C and 0.2C), the polysulfide shuttle phenomenon in the molten salt is much less than that in the ionic liquid (EMIC-AlCl₃) due to high solubility of polysulfides in ionic liquid (Fig. 3a).

As suggested by the reviewer, we have proposed some possible directions to improve the CE of molten-salt Al-S battery. Manipulating the inter-component ratios of the chloroaluminate melts, high-surface-area/catalytic host design and separator modification are promising to further improve the CE. **We have added the corresponding discussions in our revised manuscript.**

Figure R3. Characterizations of the Al anode in molten salt Al-S battery after cycling at 0.2C and 85 °C. (a) SEM images; (b-e) SEM image and corresponding elemental mappings; (f) XPS full spectrum.

6. The electrolyte and electrode design are important for metal-S batteries. Some relevant work published recently may be referred such as Nature Communications 2021, 12, 7195; Nature Communications 2021, 12, 5714.

Our response:

We appreciate the reviewer's valuable suggestion. We have now properly referred these important publications where we discuss the electrolyte and electrode design in the revised manuscript.

Reviewer #2 (Remarks to the Author):

In this work, Meng et al. report a rapid-charging Al-S battery operated at a sub-water-boiling temperature of 85 °C with a tamed quaternary molten salt electrolyte. The manuscript lacks novelty, the analysis of the results is not profound enough, and the authors do not offer any constructive insights. This manuscript is not suitable for publication in Nature Communications. Some specific comments are below.

Our response:

We thank the reviewer for raising concerns and questions on our work. We think the reviewer is concerned on two aspects: the novelty, and the analysis of the result (scientific insight). Our responses to the two points are described below as a response to the reviewer's comments.

→ On the first aspect, we would like to clarify the novelty of our manuscript by re-summarizing and rephrasing the key points in terms of molten salt electrolyte, electrochemical performance, and reaction mechanism. The detailed discussions are given as follows, and we hope that by rephrasing these statements, we can convince the reviewer of the novelty of our work.

First, we demonstrate a rapidly charging aluminium-sulfur (Al-S) battery at a sub-water-boiling temperature of 85 °C; to make the cell work at a low temperature is of great practical significance and therefore, our work of achieving this goal by designing a new quaternary alkali chloroaluminate electrolyte bears great novelty. As we know, molten salts have been granted with a sign of overly high temperature 110~600 °C and being impractical for batteries. Nevertheless, maintaining such temperature may require sophisticated sealing and thermal management systems, and restrict its mobile applications. In great contrast, with the operation temperature of our quaternary melt below 100 °C, water can be used as a low-cost heating media, and the battery start-up heating will be less energy-intensive.

From the perspective of scientific insights, of significance, we show that the quaternary melt – while exhibiting a low melting point owed to increased clustering configuration entropy, still possessing abundant electrochemically active high-order Al-Cl clusters – facilitates fast Al³⁺ desolvation at 85 °C. This has thus allowed the cell to function with a rather high rate capability at a lowered temperature. We believe that our work offers a good example to accelerate practical applications of molten-salt Al-S batteries.

Second, our findings show that the aluminum-sulfur chemistry undergoes asymmetric sulfur reaction mechanism that involves formation of polysulfide intermediates and aluminum sulfide upon discharge, as unambiguously revealed by *operando* X-ray absorption spectroscopy. Such reaction mechanism accounts for the fundamentally high reaction kinetics at such a low temperature. We performed rigorous mathematical analyses on the spectra to reveal such mechanism. In detail, during discharge, the sulfur experiences a two-phase conversion reaction in the SOD of 0-50% and a three-phase conversion process in the SOD of 50-100%, finally forming Al₂S₃ product; while during charge, the Al₂S₃ converts into the S⁰ through a three-phase conversion

process. The mechanism is apparently different than that occurs in Li-S or Na-S batteries. To our best knowledge, this is the first time to discover the asymmetric sulfur reaction mechanism for Al-S batteries.

Third, with respect to the sulfur cathode design working at a low temperature, it is desired to catalyze and facilitate the sulfur reaction; and we learned along the way that most typical oxide/sulfide catalysts are unstable in our molten salt. We have thus developed a nitrogen-functionalized porous carbon, critically being chemical compatible with the active melt, further mediates the sulfur reaction, enabling the battery with rapid-charging capability and excellent cycling stability with 85.4% capacity retention over 1400 cycles. We believe the findings here bear great scientific insights. In detail, our demonstrated Al-S battery presents a high capacity of 931 mAh g⁻¹ with a small voltage hysteresis (0.19 V) at a charging rate of C/5, and an acceptable capacity of 426 mAh g⁻¹ at a charging rate of 2C. *Such long lifetime and low polarization have not been demonstrated by any previous reports for Al-S batteries.*

→ On the second aspect, we respectfully disagree with the reviewer on commenting the analysis of the results and constructive insights, given the following facts and arguments.

First, we have systematically explored the solvation structures and desolvation kinetics of Al³⁺ at the lower operation temperature of 85 °C for the quaternary melt and traditional ionic liquid by means of spectroscopy and theoretical calculations. We learned that the solvation structures of the two specific electrolytes determine their reaction kinetics. We performed *ab initio* molecular dynamics simulations to further reveal the solvation structures around Al³⁺ of the quaternary melt and ionic liquid. The major difference between them is that the quaternary melt possesses high-order Al₃Cl₁₀⁻ along with neutral Al₂Cl₆, which are more energetically favorable for Al³⁺ desolvation. We confirmed the desolvation energetics by the calculated charge density differences of different Al-containing clusters. Through these in-depth theoretical calculations, we confirmed that the presence of abundant high-order Al_nCl_{3n+1}⁻ clusters in the quaternary melt even at a low temperature of 85 °C, which results in a low Al³⁺-desolvation barrier, laying the ground for fast cell charging.

Second, to reveal the sulfur reaction pathway of Al-S battery during cycling, *operando* X-ray absorption near-edge spectroscopy was performed to explore local structural changes of sulfur during the first cycle. Using the multivariate curve resolution-alternating least-squares (MCR-ALS) method, we de-convolved the ensemble of spectra on discharge and charge. We identified three probable primary components that mathematically describes all spectra, which we ascribe to S⁰, Al₂(S_n)₃ or Al₂(S_m)₃ intermediate, and Al₂S₃. We further observed that the Al₂(S_n)₃ intermediate on discharge is different from the Al₂(S_m)₃ intermediate on charge, indicating the asymmetric reaction mechanism of molten salt Al-S battery over discharge/charge. Further, we have performed linear combination fitting analysis of the XANES spectra using the three identified components as standards to reveal the dynamics of sulfur reactions. Therefore, *operando* X-ray absorption near-edge spectroscopy provides deep insights into this asymmetric sulfur reaction mechanism, which can give good directions for rational design of high-performance Al-S batteries.

Third, in the revised manuscript, we have made a comprehensive comparative analysis on the physical properties between our quaternary electrolytes and binary/ternary counterparts. The

physical properties including melting point, viscosity and ionic conductivity have been discussed in detail. The melting point as one of the most important physical property of molten salts has analyzed as an example, which directly determines the battery operation temperature. For molten salts, the latent heat of fusion (ΔH_f) dictates the energy required to overcome the electrostatic attraction forces between cations and anions; a weaker electrostatic attraction force leads to a lower fusion temperature. The magnitude of ΔH_f highly depends upon the intermolecular forces involved in crystal binding and evidently varies for different types of crystals. In addition, the variation of the fusion entropy (ΔS) varies for nearly all substances. Because the Gibbs free energy of a specific molten salt system is constant during fusion, the fusion temperature is determined by the basic relation of thermodynamics: $T_f = \Delta H_f / \Delta S$. On basis of the above equation, the increase of the system entropy contributes to the decrease the fusion temperature. Five principal mechanisms are classified to increase the system entropy of molten salts as follows: (a) increase in vibrational entropy; (b) increase of orientational randomization; (c) increase in positional disorder; (d) randomization of the internal configuration of molecules or ions containing groups; (e) changes of association or chemical bonding on melting. From the perspective of thermodynamics, in our quaternary melt, the presence of multiple components greatly increases the fusion entropy, leading to decreased fusion temperature.

In brief, these in-depth understandings provide constructive insights into electrolyte design and electrode design towards high-performance Al-S batteries in near future researches. We hope that by these revised arguments, the reviewer would share the same enthusiasm with us on the constructive insights of this manuscript.

We have now revised and improved some of the arguments in the revised manuscript.

1. Some similar publications related to the concept of alkali chloroaluminate melt have been reported for Al-S batteries (e.g., Nature Comm. 2021, 12(1): 5714, Energy Environ. Sci., 2022, 15, 5229-5239). What are the differences and innovations in the current manuscript compared to other published articles like these?

Our response:

We appreciate the reviewer raised the concern on the differences and innovations of our manuscript. With great enthusiasm, we have now carefully read the two publications listed by the reviewer. We respectfully disagree with the reviewer's comment on similar publications with our manuscript. Compared with the published articles, there are three major differences and innovations in our manuscript as follows:

First, the electrolyte and/or electrode materials studied in the two articles are different from those used in our study. The first article (Nature Comm. 2021, 12(1), 5714) studied organic ionic liquid electrolytes, which consists of $AlCl_3$ and EMIC and operates at room temperature. In contrast, our work reports the finding of using an inorganic alkali chloroaluminate melt electrolyte, which is different from ionic liquid in terms of the formation of active Al_xCl_y clusters, the reaction kinetics and mass transport properties (as discussed below). The second article (Energy Environ. Sci., 2022, 15, 5229-5239) reports on a battery that use activated carbon as both the anode and

cathode, which functions as an electrochemical capacitor, and is a completely different electrochemical system than our study (aluminium-sulfur batteries). Also, this work reports on the use of a ternary melt (consisting of AlCl_3 , NaCl and LiCl); in contrast, our work designed a specific quaternary melt that exhibits a low melting point owed to increased clustering configuration entropy.

Second, we identified a different sulfur reaction mechanism than what is found in organic ionic liquids. The first article reports an aluminum-sulfur battery that is based on a reversible sulfur oxidation process in ionic liquid, where sulfur is electrochemically oxidized to form AlSCl_7 (S^{4+}). In our manuscript, the sulfur conversion is between S_0 and Al_2S_3 , which exploits different oxidation states of the sulfur. Our findings show that the asymmetric sulfur reaction mechanism that involves formation of polysulfide intermediates, as revealed by *operando* X-ray absorption spectroscopy, accounts for the high reaction kinetics at 85 °C. During discharge, the sulfur experiences a two-phase conversion reaction in the SOD of 0-50% and a three-phase conversion process in the SOD of 50-100%, finally forming Al_2S_3 product; while during charge, the Al_2S_3 converts into the S^0 through a three-phase conversion process. For the second article, the activated carbon exhibits a physisorption-chemisorption based capacitive storage mechanism under electrolyte nanoconfinement. Therefore, our manuscript is totally different from other two articles in terms of battery reaction mechanisms.

Third, our manuscript demonstrates improved battery performances in terms of capacity, rate capability and cycling stability. In our manuscript, the aluminum-sulfur battery showed rapid-charging capability and excellent cycling stability with 85.4% capacity retention over 1400 cycles. In detail, it presents a high capacity of 931 mAh g^{-1} with a small voltage hysteresis (0.19 V) at a charging rate of C/5, and a rather high capacity of 426 mAh g^{-1} at a charging rate of 2 C. Such long lifetime and low polarization have not been demonstrated by any previous reports for Al-S batteries. In the first article, the Al-S battery nicely exhibits a high operation voltage of ~1.8 V, but shows a relatively low specific capacity (~225 mAh g^{-1}), fast capacity fading and rate performance. In the second article, the activated carbon displays excellent rate performance and good cycling stability due to its physisorption-chemisorption mechanism, but shows a very low specific capacity (~116 mAh g^{-1}).

Taken together, we believe our manuscript shows major differences and innovations compared to other articles. We have now reiterated the novelty of our manuscript in terms of molten salt electrolyte, electrochemical performance, and reaction mechanism in the revised manuscript.

2. The authors claimed that even though the dissolution of aluminum polysulfides is detected, it does not lead to capacity fading since the polysulfide species do not form inactive solids on Al surface at the anode over the course of shuttling. However, this claim lacks supporting characterization of the Al-anode. Furthermore, the long-term cycling performance at C/5 and 1C reveals a low coulombic efficiency of 85%. Do the authors provide an explanation for the root cause of this low coulombic efficiency?

Our response:

We appreciate the reviewer's great suggestion on the dissolution of aluminum polysulfide and the Coulombic efficiency. This is a very constructive comment and has strengthened our argument. We have made performed further characterization of the aluminum anode and made further discussions on the Coulombic efficiency.

To answer the question on the root cause of the average Coulombic efficiency of ~84%, we propose that this is due to the dissolution of polysulfide intermediates in the melt electrolyte and thus polysulfide shuttle during charge. We have observed the color darkening of the electrolyte after cycling, which provide the hint of polysulfide dissolution.

Further, as suggested by the reviewer, to confirm that the partially dissolved aluminium polysulfides do not form inactive solids on Al surface over the course of shuttling, we investigated the Al anode surface after cycling the cell at a rate of 0.2 C and 85 °C. The resulting Al anode was fully washed by hexane to remove residual molten salt. The SEM image shows compact micron-sized crystals on the Al surface (Fig. R3a). The imaging and the corresponding elemental mappings indicate that the Al anode is sulfur-free (at least to the limits of detection of energy-dispersive X-ray spectroscopy), indicating the absence of sulfur-containing particles (Fig. R3b-e). In addition, high-resolution XPS measurement was performed to explore the surface information of the Al anode. There are no sulfur signals in the XPS full spectrum (Fig. R3f). Therefore, these results confirm the absence of inactive polysulfide solids on Al surface in molten salt Al-S battery after cycling. **We have added the corresponding discussions in our revised manuscript.**

Figure R3. Characterizations of the Al anode in molten salt Al-S battery after cycling at 0.2C and 85 °C. (a) SEM images; (b-e) SEM image and corresponding elemental mappings; (f) XPS full spectrum.

3. Ionic conductivity is one of the important parameters to evaluate the motion of ionic charge in the electrolyte. Have the authors tried measuring the conductivity of the alkali chloroaluminate melt electrolytes?

Our response:

We appreciate the reviewer's valuable suggestion. As kindly suggested by the reviewer, we have now carried out the ionic conductivity measurement of our quaternary alkali chloroaluminate melt electrolyte using a Mo|Mo blocking cell configuration with a glass fiber separator at 85 °C. We calculated the ionic conductivity of the electrolytes based on electrochemical impedance spectroscopy measurement by analyzing the Nyquist plot. The ionic conductivity (σ) is calculated by the equation: $\sigma = d/(R_b * A)$; where d is the distance between the two electrodes, and A is the geometric area of the electrolyte electrode, R_b is the ohmic resistance attributed to the electrolyte. In our configuration cell, d is determined by the thickness of glass fiber and valued as 0.06 cm, and A is 1.13 cm². As shown in Figure R2, the R_b value is 8.1 Ω . Therefore, the ionic conductivity is calculated to be around 0.66 S/m. We have added the corresponding discussions in our revised manuscript.

Figure R4. The Nyquist plot of the Mo-Mo cell with LiCl-NaCl-KCl-AlCl₃ at 85 °C.

4. The authors mentioned that there is a dendrite suppression effect in the quaternary melt electrolyte based on the Al|Al symmetric cell results. Is there any proposed mechanism and experimental result to support this?

Our response:

We appreciate the reviewer's valuable comment. We have now performed further experiments and made discussions on the dendrite suppression effect of the quaternary melt. We conducted morphological studies on Al plating by comparing the two electrolytes (alkali melt and ionic liquid) at the same current density of 1 mA cm⁻² with an areal capacity of 3 mAh cm⁻². In the ionic liquid electrolyte (EMIC-AlCl₃), the Al deposits show the presence of many tiny particles with size below 2 μ m and large aggregates (Fig. R5a). The corresponding EDS elemental mapping image confirms the existence of only Al deposits on the Mo substrate (Fig. R5b). In great contrast, the Al deposits in the quaternary melt electrolyte (LiCl-NaCl-KCl-AlCl₃) exhibit compact micron-sized crystals

with well-defined facets and relatively large size above 20 μm (Fig. R5c, d). These large crystals confirm that the aluminum plating in the molten salt does not follow a dendrite-like growth mechanism, but rather bulk particle-like mechanism. On basis of the typical heterogeneous nucleation theory, we propose that the obvious difference in the two electrolytes is attributed to lower surface tension between nucleus and electrolyte, and faster mass diffusion kinetics in the quaternary melt. We have added the corresponding discussions in our revised manuscript.

Figure R5. Characterizations of the plated Al on Mo substrate in the Al-Mo cells using ionic liquid electrolyte and quaternary alkali chloroaluminate melt electrolyte. a, b SEM image and elemental mapping image of the plated Al on Mo substrate using ionic liquid electrolyte; c, d SEM image and elemental mapping image of the plated Al on Mo substrate using quaternary melt electrolyte.

Furthermore, we have now also performed time-of-flight secondary ion mass spectrometry (TOF-SIMS) to identify the chemical compositions near the surface of the Al anode after cycling in the ionic liquid and quaternary alkali melt electrolytes. We investigated the depth distribution by using ion beam to remove layers from the anode surface while analyzing the mass spectra of the surface (Fig. R6). For the ionic liquid electrolyte, the C^+ , AlCl^+ and CH_2^+ secondary ions show a large presence on the surface, and an obvious decrease along with the increase of depth,

indicating that a large fraction of these secondary ions on the outer surface. For example, a 3D depth rendering image of C^+ confirms that the carbon component is enriched in the outer surface. This indicates the formation of the solid electrolyte interphase (SEI) on the Al surface with the ionic liquid electrolyte, which consists of a mixture of organic and inorganic components. In great contrast, with the quaternary alkali melt electrolyte, the intensity of C^+ , $AlCl^+$ and Cl^+ is about two orders of magnitude lower than that observed for the ionic liquids; and no CH_2^+ is observed. This indicates that as none organic components are present, the alkali melt does not undergo reactions with aluminum metal and no SEI is observed. This difference in the interphase between the two electrolytes is confirmed by 2D chemical ion images (Figs. R7 and R8). We therefore propose that the absence of SEI components on the Al anode surface also leads to non-dendritic growth in the quaternary alkali chloroaluminate melt electrolyte. We have added the corresponding discussions in our revised manuscript.

Figure R6. TOF-SIMS of the Al anode surface after cycling in the ionic liquid and quaternary melt electrolytes. (a) Depth profiles of different secondary ions in the ionic liquid electrolyte. (b, c) 3D images of the sputtered volume corresponding to the depth profiles in the ionic liquid electrolyte. (d) Depth profiles of different secondary ions in the quaternary melt electrolyte. (e, f) 3D images of the sputtered volume corresponding to the depth profiles in the quaternary melt electrolyte.

Figure R7. TOF-SIMS 2D images of different secondary ions on the Al anode surface after cycling in the ionic liquid electrolyte.

Figure R8. TOF-SIMS 2D images of different secondary ions on the Al anode surface after cycling in the quaternary melt electrolyte.

5. What is the function of KCl?

Our response:

We appreciate the reviewer's valuable comment. In our manuscript, the main function of KCl is to act as an important alkali metal salt and thus form quaternary alkali chloroaluminate melt electrolyte by coupling with LiCl, NaCl and AlCl₃. As we discussed above, we have made a comprehensive comparative analysis on the physical properties between our quaternary electrolytes and binary/ternary counterparts.

As one of the most important physical property of molten salts, the melting point as an example is discussed in detail, which directly determines the battery operation temperature. For molten salts, the latent heat of fusion (ΔH_f) dictates the energy required to overcome the electrostatic attraction forces between cations and anions; a weaker electrostatic attraction force leads to a lower fusion temperature. The magnitude of ΔH_f highly depends upon the intermolecular forces involved in crystal binding and evidently varies for different types of salts. In addition, the variation of the fusion entropy (ΔS) plays great role. Because the Gibbs free energy of a specific molten salt system is constant during fusion, the fusion temperature is determined by the basic relation of thermodynamics: $T_f = \Delta H_f / \Delta S$. On basis of the above equation, the increase of the entropy of the system contributes to the decrease the fusion temperature. Five principal mechanisms are classified to increase the system entropy of molten salts as follows: (a) increase in vibrational entropy; (b) increase of orientational randomization; (c) increase in positional disorder; (d) randomization of the internal configuration of molecules or ions containing groups; (e) changes of association or chemical bonding on melting. From the perspective of thermodynamics, in our quaternary melt, the presence of multiple components greatly increases the fusion entropy, leading to decreased fusion temperature. As shown in Table S1, we can observe that the quaternary molten salts generally show much lower melting point than single, binary and ternary components, further confirming the significance of multiple components in molten salts.

6. The authors claim that the energy density of Al-S batteries is moderate; however, the areal loading of sulfur is only 1.5 mg cm⁻². To explore the practical feasibility further, has the molten salt electrolyte system been tested with high sulfur loading in the range of 3 to 5 mg cm⁻²?

Our response:

We appreciate the reviewer's very constructive suggestion. We have now investigated the electrochemical performance of the molten salt Al-S battery at a high sulfur loading. As shown in Fig. R9, the electrode with a sulfur loading of 3.1 mg cm⁻² exhibits a high reversible capacity of ~760 mAh g⁻¹ and good cycling stability at 85 °C and 0.2C. Even at a high sulfur loading of 6.5 mg cm⁻² and 0.1C, the electrode shows an obvious capacity increase up to 530 mAh g⁻¹ after 40 cycles.

To achieve higher cycling performance of Al-S battery at a higher mass loading, we advocate that further efforts on cathode design and electrolyte formulations are required. For example, rational design of 3D interconnected porous sulfur/carbon composite electrode with bicontinuous ion and electron transports may facilitate electrolyte penetration and mass transport in thick electrodes. **We have now added the data and discussed this thoroughly in the manuscript.**

Figure R9. Electrochemical performance of the Al-S battery using the quaternary molten salt electrolyte at high sulfur loading. (a, b) Cycling performance and voltage profiles of the S electrode with a mass loading of 3.1 mg cm⁻² at 0.2C and 85 °C. (c, d) Cycling performance and voltage profiles of the S electrode with a mass loading of 6.5 mg cm⁻² at 0.1C and 85 °C.

Reviewer #3 (Remarks to the Author):

This manuscript reports the utilization of a quaternary molten salt electrolyte for Al-S battery, which exhibit desirable high reversible capacity as well as outstanding cycling stability. The structure of the molten electrolyte as well as the charge/discharge working mechanism of the Al-S battery has been investigated with rational characterizations. The manuscript has been carried out with logical writing and well-supported conclusions. Therefore, I recommend the manuscript to be considered for publication after minor revisions as follows:

Our response: We fully appreciate the reviewer's thoughtful and encouraging comments about our manuscript, and offering the opportunity to address and clarify the issues raised in the report. Our responses to the points raised in the report are described below following the reviewer's comments.

1. The melting points of the individual salts, as well as the binary and ternary salts should be supplemented for reference.

Our response:

We appreciate the reviewer's valuable comment.

We have now added typical phase diagrams and the melting points of different chloride salt in our revised supporting information.

Figure R1. Typical phase diagrams of binary and ternary molten salt systems.

Table R1. A summary on the chloride salt components, molar ratios and melting points of different molten salts.

Molten salts	Salt components	Molar ratios	Melting point (°C)
Single	LiCl	1	605
	NaCl	1	801
	KCl	1	858
	AlCl ₃	1	190
Binary	LiCl/NaCl	72/28	549
	KCl/NaCl	50/50	652
	KCl/LiCl	42/58	354
	AlCl ₃ /LiCl	58/42	110
	AlCl ₃ /NaCl	61/39	102
	AlCl ₃ /KCl	66/34	128
Ternary	AlCl ₃ /NaCl/KCl	61/26/13	~100
	AlCl ₃ /LiCl/KCl	59/29/12	~95
Quaternary	AlCl ₃ /LiCl/NaCl/KCl	120/42/43/15	~80

2. Previous reports have demonstrated the utilization of alkali metal chlorides as additives for ionic liquid electrolyte to modify the interface stability of Al anodes. Does the quaternary salt have similar functions? The compositions of the interfaces on the Al anodes should be investigated.

Our response:

We appreciate the reviewer's valuable suggestion. We totally agree with the reviewer's comment on the role of alkali metal chlorides as additives. As demonstrated in previous reports (J. Electrochem. Soc., 2005, 152, 620-625; Electrochem., 2005, 73, 739-741; Adv. Funct. Mater., 2023, 33, 2214405), alkali metal chlorides not only improve the ionic conductivity of the ionic liquid electrolyte, but also facilitate the desolvation process by forming relatively strong interaction between chloride ions with alkali metal ions. Therefore, the alkali metal chlorides in our quaternary melt electrolyte have similar functions, which are beneficial for the interface stability of Al anodes during cycling.

Furthermore, we have now also performed time-of-flight secondary ion mass spectrometry (TOF-SIMS) to identify the chemical compositions near the surface of the Al anode after cycling in the ionic liquid and quaternary alkali melt electrolytes. We investigated the depth distribution by using ion beam to remove layers from the anode surface while analyzing the mass spectra of the surface (Fig. R6). For the ionic liquid electrolyte, the C^+ , $AlCl^+$ and CH_2^+ secondary ions show a large presence on the surface, and an obvious decrease along with the increase of depth, indicating that a large fraction of these secondary ions on the outer surface. For example, a 3D depth rendering image of C^+ confirms that the carbon component is enriched in the outer surface. This indicates the formation of the solid electrolyte interphase (SEI) on the Al surface with the ionic liquid electrolyte, which consists of a mixture of organic and inorganic components. In great contrast, with the quaternary alkali melt electrolyte, the intensity of C^+ , $AlCl^+$ and Cl^+ is about two orders of magnitude lower than that observed for the ionic liquids; and no CH_2^+ is observed. This indicates that as none organic components are present, the alkali melt does not undergo reactions with aluminum metal and no SEI is observed. This difference in the interphase between the two electrolytes is confirmed by 2D chemical ion images (Figs. R7 and R8). We therefore propose that the absence of SEI components on the Al anode surface also leads to non-dendritic growth in the quaternary alkali chloroaluminate melt electrolyte. We have added the corresponding discussions in our revised manuscript.

Figure R6. TOF-SIMS of the Al anode surface after cycling in the ionic liquid and quaternary melt electrolytes. (a) Depth profiles of different secondary ions in the ionic liquid electrolyte. (b, c) 3D images of the sputtered volume corresponding to the depth profiles in the ionic liquid electrolyte. (d) Depth profiles of different secondary ions in the quaternary melt

electrolyte. (e, f) 3D images of the sputtered volume corresponding to the depth profiles in the quaternary melt electrolyte.

Figure R7. TOF-SIMS 2D images of different secondary ions on the Al anode surface after cycling in the ionic liquid electrolyte.

Figure R8. TOF-SIMS 2D images of different secondary ions on the Al anode surface after cycling in the quaternary melt electrolyte.

3. The authors have published results for Al-chalcogen batteries with the ternary NaCl-KCl-AlCl₃ salt in Nature 608, 704–711 (2022), therefore the necessity of the LiCl in the quaternary salt should be discussed.

Our response:

We appreciate the reviewer's valuable suggestion. In our manuscript, the main necessity of LiCl is to act as an important alkali metal salt and thus form quaternary alkali chloroaluminate melt electrolyte by coupling with NaCl, KCl and AlCl₃. As we discussed above, we have made a comprehensive comparative analysis on the physical properties between our quaternary electrolytes and binary/ternary counterparts.

As one of the most important physical property of molten salts, the melting point as an example is discussed in detail, which directly determines the battery operation temperature. For molten salts, the latent heat of fusion (ΔH_f) dictates the energy required to overcome the electrostatic attraction forces between cations and anions; a weaker electrostatic attraction force leads to a lower fusion temperature. The magnitude of ΔH_f highly depends upon the intermolecular forces involved in crystal binding and evidently varies for different types of crystals. In addition, the variation of the fusion entropy (ΔS) plays great role. Because the Gibbs free energy of a specific molten salt system is constant during fusion, the fusion temperature is determined by the basic relation of thermodynamics: $T_f = \Delta H_f / \Delta S$. On basis of the above equation, the increase of the system entropy contributes to the decrease the fusion temperature. Five principal mechanisms are classified to increase the system entropy of molten salts as follows: (a) increase in vibrational entropy; (b) increase of orientational randomization; (c) increase in positional disorder; (d) randomization of the internal configuration of molecules or ions containing groups; (e) changes of association or chemical bonding on melting. From the perspective of thermodynamics, in our quaternary melt, the presence of multiple components greatly increases the fusion entropy, leading to decreased fusion temperature. As shown in Table S1, we can observe that the quaternary molten salts generally show much lower melting point than single, binary and ternary components, further confirming the significance of multiple components in molten salts. Besides, the quaternary melt shows great advantages in terms of ionic conductivity and viscosity.

4. The thickness of the Al foil and the dosage of electrolyte in each cell should be specified.

Our response:

We appreciate the reviewer's valuable suggestion. We have added these parameters in our revised manuscript.

REVIEWERS' COMMENTS

Reviewer #2 (Remarks to the Author):

I have carefully evaluated the response of the authors to my comments and the necessary revisions made in the manuscript. As outlined below, the following points still need to be clarified before it can be considered for publication in Nature Communications.

1. When the operation temperature increases to 85 °C, the voltage window seems to reduce (Figure 3a). The authors should explain why different voltage windows were used at different temperatures.

3. Following my previous question 6, the authors investigated the high S loading performance. However, there is a very strange phenomenon in Supplementary Fig. 14c. Why does the capacity slowly increase in the case of 6.5 mg cm⁻² loading, while in other cells it does not happen? Therefore, would the capacity of the cell keep increasing and when would it begin to decay? Can the authors account for this and explain?

3. Following my previous question 2, the authors confirmed that the dissolution of polysulfide intermediates in the melt electrolyte and thus polysulfide shuttle during charge process. However, the cycle performance of the cell at 0.5C and 1C rates (Figure 3d and 3g) still seems to be very stable. If there is a shuttle effect, then the cell capacity should have obviously a decay phenomenon. Can the authors explain and account for it?

Reviewer #2 (Remarks to the Author):

I have carefully evaluated the response of the authors to my comments and the necessary revisions made in the manuscript. As outlined below, the following points still need to be clarified before it can be considered for publication in Nature Communications.

Our response: We fully appreciate the reviewer's thoughtful comments about our manuscript, and offering the opportunity to address and clarify the issues raised in the report. Our responses to the points raised in the report are described below following the reviewer's comments.

1. When the operation temperature increases to 85 °C, the voltage window seems to reduce (Figure 3a). The authors should explain why different voltage windows were used at different temperatures.

Our response:

We appreciate the reviewer's comment. In Figure 3a, we present the voltage profiles of Al-S cells with a S/CNF cathode using the quaternary melt and ionic liquid electrolytes at different operation temperatures. The used voltage range depends on the voltage polarization and hence the exact redox potential of the cathode under different reaction conditions. In general, we make an effort to use a narrow voltage window (a window enough so as to allow the sulfur reaction to complete). At an operating temperature of 85 °C, the two systems, EMIC-AlCl₃ and quaternary alkali melt-based Al-S cells were compared with the same voltage window and given that the reaction at this temperature poses rather low polarization, we used a small window of 0.5-1.3 V. However, at room temperature (25 °C), the sulfur conversion reaction kinetics in ionic liquid is very sluggish, and therefore we expanded the voltage window to 0.2-1.6 V. At a rate of 0.2 C and 25 °C, the EMIC-AlCl₃ based Al-S cell using a S/NCF cathode shows a large voltage polarization of approximately 1.06 V. Therefore, This substantial difference in sulfur conversion kinetics at different temperatures (e.g., 85 °C versus 25 °C) necessitates the use of different voltage windows. We have added the corresponding discussions in our revised manuscript.

2. Following my previous question 6, the authors investigated the high S loading performance. However, there is a very strange phenomenon in Supplementary Fig. 14c. Why does the capacity slowly increase in the case of 6.5 mg cm⁻² loading, while in other cells it does not happen? Therefore, would the capacity of the cell keep increasing and when would it begin to decay? Can the authors account for this and explain?

Our response:

We appreciate the reviewer's valuable suggestion. Indeed, we have observed an increase in capacity in the case of a high sulfur loading of 6.5 mg cm^{-2} . It is worth noting that this capacity increase phenomenon is commonly observed in high-loading sulfur electrodes in traditional Li-S batteries (ACS Nano 2016, 10, 10462-1047; J. Mater. Chem. A, 2020, 8, 3027-3034; Adv. Energy Mater. 2022, 12, 2201585.). In our Al-S battery system, this phenomenon is likely attributed to the sluggish electrochemical reactions involving Al^{3+} participation.

We have now extensively discussed the capacity increase phenomenon of high-loading sulfur electrodes in our revised manuscript. This phenomenon can be attributed to two factors as follows. **One important factor lies on the nature of the thick sulfur electrode.** To obtain a higher mass loading of the sulfur electrode, we prepared a thick and dense S/CNF composite cathode using a slurry casting method. However, the tortuous pore structures in thick composite electrodes makes it challenging for the quaternary alkali melt to fully infiltrate the entire electrode in short time (during the cell's resting and for the first few cycles). Therefore, we speculate that during the initial cycle, the conversion reaction of the thick S/CNF cathode primarily occurs on the outer electrode layer, while the inner layer gradually becomes wetted with the electrolyte and participates in the electrochemical reaction as the cycling progresses. **Another factor lies on the quaternary alkali melt with relatively high viscosity at the operation temperature of $85 \text{ }^\circ\text{C}$.** As mentioned in our manuscript, the quaternary alkali chloroaluminate melt based on an $\text{AlCl}_3\text{-NaCl-LiCl-KCl}$ mixture has a lower melting point of approximately $80 \text{ }^\circ\text{C}$. Since the cell's operating temperature only slightly exceeds the melting point of the electrolyte, we speculate that the quaternary alkali melt exhibits relatively high viscosity, resulting in insufficient electrolyte penetration into the thick electrode. Based on these discussions, the capacity of the Al-S cell will not continue to increase when the electrode is fully wetted by the electrolyte during cycling. The cell may decay when some of the irreversibility accumulates and causes loss of sulfur in the electrolyte.

To achieve higher cycling performance of Al-S batteries with higher mass loadings, we emphasize the need for further advancements in cathode design and electrolyte formulations. One approach is the rational design of 3D interconnected porous sulfur/carbon composite electrodes with bicontinuous ion and electron transport. This design can facilitate electrolyte penetration and enhance mass transport within thick electrodes, allowing for good accommodation of the large volume changes resulting from sulfur conversion reactions during cycling. Previous studies have reported well-designed high-loading sulfur electrodes in high-energy Li-S batteries, which can serve as valuable references (Matter 2020, 2, 1605-1620; Nano Lett. 2022, 22, 5982-5989; Nat. Commun. 2021, 12, 4519; Adv. Mater. 2016, 28, 3374-3382.). **We have added the corresponding discussions in our revised manuscript.**

3. Following my previous question 2, the authors confirmed that the dissolution of

polysulfide intermediates in the melt electrolyte and thus polysulfide shuttle during charge process. However, the cycle performance of the cell at 0.5C and 1C rates (Figure 3d and 3g) still seems to be very stable. If there is a shuttle effect, then the cell capacity should have obviously a decay phenomenon. Can the authors explain and account for it?

Our response:

Thank you for pointing out the question, and we would like to further clarify the relationship between the shuttle effect and capacity decay. We have addressed this concern in our revised manuscript, which has greatly strengthened the argument thanks to the reviewer's question.

As shown in Figure 3d and 3g, we observed that the polysulfide shuttle phenomenon is still present in the molten salt, but importantly, it does not lead to capacity fading as typically observed in Li-S batteries. We would like to first note that polysulfide shuttling does not by nature lead to capacity fading. The capacity fading in Li-S due to polysulfide shuttling is only because the formed low-order polysulfides (Li_2S_x) are solid and therefore remained on the lithium anode and thus are permanently inactive. Based on this principle, we speculate that this non-fading capacity phenomenon occurs because there is none formation of solid, irreversible products by reaction of polysulfide with the aluminium anode in the inorganic molten salt during cycling. In another words, the dissolved polysulfide may diffuse to and react with the Al anode, but the formed products are not solid, but rather soluble in the molten salt, and then fully diffuse back to the cathode to participate in further oxidation reactions. This means polysulfide shuttling occurs but there is no continuous loss of sulfur to the anode surface.

We have conducted a series of characterizations to confirm the absence of inactive polysulfide solids on the Al surface in the molten salt Al-S battery after cycling. In detail, the SEM image shows compact micron-sized crystals on the Al surface (Supplementary Fig. 13a). The imaging and the corresponding elemental mappings indicate that the Al anode is sulfur-free (at least to the limits of detection of energy-dispersive X-ray spectroscopy), indicating the absence of sulfur-containing particles (Supplementary Fig. 13b-e). In addition, high-resolution XPS measurement was performed to explore the surface information of the Al anode. There are no sulfur signals in the XPS full spectrum (Supplementary Fig. 13f). Moreover, the absence of SEI components on the Al anode surface was confirmed through TOF-SIMS analysis (Supplementary Figs. 5-7). With the quaternary alkali melt electrolyte, the intensity of C^+ , AlCl^+ and Cl^+ is about two orders of magnitude lower than that observed for the ionic liquids; and no CH_2^+ is observed. This indicates that as none organic components are present, the alkali melt does not undergo reactions with aluminum metal and no SEI is observed.

In brief, the absence of inactive polysulfide solids and SEI on the Al surface in molten salt Al-S batteries fundamentally distinguishes their electrochemistry from the conventional understanding of sulfur-based systems. We propose that this absence of inactive species on the Al surface is crucial for the high cycling stability observed in molten

salt Al-S batteries. These findings have been incorporated into our revised manuscript to provide a clear explanation of the relationship between the shuttle effect and capacity decay in our system.

Further efforts in tuning the chloroaluminate melt chemistry, designing high-surface-area/catalytic hosts and modifying separators are expected to allow further improvement on the CE. **We have added the corresponding discussions in our revised manuscript.**